# Habenula contributions to negative self-cognitions

Po-Han Kung [1,2], Matthew D. Greaves [1,3], Eva Guerrero-Hreins [4,5], Ben J. Harrison[2], Christopher G. Davey [2], Kim L. Felmingham[1], Holly Carey[2], Priya Sumithran[6,7], Robyn M. Brown [4,5], Bradford A. Moffat [8], Rebecca K. Glarin[8], Alec J. Jamieson [2] & Trevor Steward [1,2] ✉

Self-related cognitions are integral to personal identity and psychological wellbeing. Persistent engagement with negative self-cognitions can precipitate mental ill health; whereas the ability to restructure them is protective. Here, we leverage ultra-high field 7T fMRI and dynamic causal modelling to characterise a negative self-cognition network centred on the habenula – a small midbrain region linked to the encoding of punishment and negative outcomes. We model habenula effective connectivity in a discovery sample of healthy young adults ($n = 45$) and in a replication cohort ($n = 56$) using a cognitive restructuring task during which participants repeated or restructured negative self-cognitions. The restructuring of negative self-cognitions elicits an excitatory effect from the habenula to the posterior orbitofrontal cortex that is reliably observed across both samples. Furthermore, we identify an excitatory effect of the habenula on the posterior cingulate cortex during both the repeating and restructuring of self-cognitions. Our study provides evidence demonstrating the habenula's contribution to processing self-cognitions. These findings yield unique insights into habenula's function beyond processing external reward/punishment to include abstract internal experiences.

Self-cognitions are thoughts and beliefs about how an individual perceives themselves, their attributes, as well as their relationship with others and the world[1]. These are often derived from personal experiences, helping to form a cohesive narrative of "Who am I?" and "What am I like?" that defines the distinctly human phenomenon of having a sense of 'self'[2]. Importantly, self-cognitions influence how one evaluates and interprets past events, responds to present circumstances, and predicts future situations[1]. As such, self-cognitions are deeply intertwined with an individual's affective experience and play a principal role in psychological

wellbeing[3,4]. For instance, persistent engagement with negative self-cognitions in the form of repetitive negative thinking has been shown to contribute to mental ill health[5,6]. In contrast, the ability to restructure and update negative self-cognitions with more adaptive narratives can alleviate negative affect and act as a protective factor for mental wellbeing[7,8]. Despite their significance to mental wellbeing, the brain mechanisms supporting the higher-order processing of self-cognitions remain largely unexplored. Understanding the neural mechanisms of negative self-cognitions would provide valuable insight into the biological

[1]Melbourne School of Psychological Sciences, Faculty of Medicine, Dentistry and Health Sciences, University of Melbourne, Victoria, Australia. [2]Department of Psychiatry, University of Melbourne, Victoria, Australia. [3]School of Psychological Sciences, Monash University, Victoria, Australia. [4]Department of Biochemistry and Pharmacology, University of Melbourne, Victoria, Australia. [5]Florey Institute of Neuroscience and Mental Health, University of Melbourne, Victoria, Australia. [6]Department of Surgery, School of Translational Medicine, Monash University, Victoria, Australia. [7]Department of Endocrinology and Diabetes, Alfred Health, Victoria, Australia. [8]Melbourne Brain Centre Imaging Unit, Department of Radiology, University of Melbourne, Victoria, Australia. ✉e-mail: trevor.steward@unimelb.edu.au

basis of maladaptive thinking patterns, such as rumination, which contribute to depression and anxiety disorders[6,9,10].

The habenula – a pair of small midbrain nuclei adjacent to the posterior mediodorsal thalamus – may play a role in encoding negative self-cognitions owing to its distinctive function in the processing of other negative stimuli[11,12]. Converging animal and human studies have shown that habenula activity increases in response to the omission of expected reward[13,14] and to the delivery of punishment[15]. Conversely, the majority of habenula neurons show reduced firing during unexpected reward receipt and when facing reward-predictive cues[16,17]. The habenula's unique function in signalling negative valence and non-reward events has led to it being recognised as the 'anti-reward' centre of the brain[18]. In particular, rodent models have shown that neurochemical activation of the habenula induces depression-like symptoms characterised by reduced mobility and sucrose preference, which can be alleviated via pharmacological inhibition of the habenula[19]. These findings have been complemented by human studies reporting increased habenula volume and activity in depression[20,21], as well as intensified habenula activity in response to negative feedback during cognitive tasks[22].

The habenula's contribution to shaping affective and behavioural responses to negative stimuli is likely underpinned by its extensive connectivity bridging the forebrain to midbrain monoamine systems[23,24]. Specifically, the habenula receives efferent projections from the medial prefrontal cortex (mPFC), the basal ganglia (e.g., globus pallidus), and the lateral hypothalamus, which supply information related to an individual's motivational state[19,25]. The habenula, in turn, modulates downstream neurotransmission to shape cognition and behaviour via bidirectional connections with the ventral tegmental area (VTA), substantia nigra compacta, and the raphe nucleus[26–28]. In rats, synaptic potentiation of habenula neurons projecting to the VTA has been found to modulate learned helplessness[29], and elevated habenula activity has been shown to induce depressive behaviours by reducing serotonin transmission from the dorsal raphe[30,31]. Moreover, signals transmitted from the habenula to the mPFC via VTA dopaminergic neurons mediate conditioned place aversion in rats[32,33]. Relatedly, interactions between habenula neurons and the anterior cingulate cortex have been shown to guide choice shifting in response to unrewarding outcomes in primates during a reversal learning task[34]. However, it is unclear if such habenula-mediated functions extend to higher-order cognition, such as negative self-related cognitions.

As the processing of negative self-cognitions is a uniquely human process, the involvement of the habenula in encoding and restructuring self-cognitions cannot be tested via animal models. To date, human neuroimaging studies have largely focused on habenula response to primary reward or punishment, such as electric shocks[15,27,35], or under task-free/resting-state conditions[36–38]. In addition to reward processing regions, resting-state imaging studies have found that habenula activity is correlated with activity of the orbitofrontal cortex (OFC), posterior cingulate cortex (PCC), and the hippocampus – key regions in networks implicated in outcome valuation, self-referential cognition, and memory functioning[36,38,39]. The OFC has been hypothesised to receive valence-based information from the habenula to compute the expected value of stimuli that guides adaptive behaviours[40]. Furthermore, the habenula is found to preferentially interface with task-positive regions of the default mode network, including the mid- to posterior-PCC, potentially suggesting a role of the habenula in self-oriented processes engaged by external activities[36]. In rats, synchronised activity between the habenula and the hippocampus is thought to contribute to the encoding of contextual information required for mnemonic processes[41,42]. However, a mechanistic account of the habenula's influence over these regions to support the processing of negative self-related cognitions remains undefined. Furthermore, assessing habenula response using standard 3-Tesla functional MRI (fMRI) is challenging as limitations in signal contrast and spatial resolution hinder the accurate delineation of the habenula from nearby structures[26,43]. Ultra-high field (7T) MRI can overcome these challenges by providing the superior image resolution and signal-to-noise ratio required to map habenula function[44–46].

In this work, we characterise the habenula's involvement in the processing of negative self-cognitions. We sought (1) to characterise habenula activity during the encoding and restructuring of negative self-cognitions and (2) to map the directional influence between the habenula and regions implicated in negative self-cognition processing via dynamic causal modelling (DCM). Under a Bayesian framework, DCM uses a neurobiologically informed generative model to infer the causal excitatory and inhibitory effects brain regions have on one another (i.e., effective connectivity), as well as to classify how these interactions are modulated by experimental tasks[47–49]. Given the habenula's role in signalling negative valence, we hypothesised that habenula activity would increase during the encoding of negative self-cognitions, and that this would be heightened when participants repeat negative cognitions as opposed to restructuring them. We further hypothesised that both the repeating and restructuring of negative self-cognitions would positively modulate connectivity within our habenula-centric network. In addition, we examined the extent to which habenula connectivity during negative self-cognition processing was associated with participants' endorsement of negative cognitions, as well as their tendency to engage in repetitive negative thinking. We carried out our analyses using a discovery sample including 48 healthy participants and tested the replicability of our obtained findings in an independent replication sample comprising 65 healthy participants.

Using 7T fMRI, we demonstrate that the repetition of negative self-cognitions elicits heightened activity in the habenula compared to restructuring, alongside regions implicated in self-directed thinking (e.g., PCC), outcome valuation (e.g., OFC), and memory (e.g., hippocampus). DCM analyses in the discovery sample reveal that the habenula exerted an excitatory influence on the PCC during both the restructuring and repeating of negative cognitions. In contrast, restructuring negative self-cognitions is characterised by the habenula having an excitatory modulatory effect on the OFC. Our replication sample corroborates this excitatory effect from the habenula to the OFC during the restructuring of negative self-cognitions, providing consistent evidence for the habenula's involvement in processing negatively valanced self-cognitions that extend beyond its previously limited role in encoding primary punishment and reward.

## Results

### Cognitive restructuring paradigm

All participants completed a block-design cognitive restructuring paradigm[50] while undergoing fMRI scanning (Fig. 1; detailed in Methods). Prior to scanning, participants received training on how to restructure negative self-cognitions using Socratic questioning techniques, such as logical reasoning and perspective shifting[51]. At the start of each block, participants were presented with a commonly reported negative self-cognition statement[52,53] and given the option to restructure or to repeat each statement. For half of the task blocks, participants restructured the negative self-cognition statements (challenge condition) using previously taught Socratic questioning techniques[51]. For the other half of the task blocks, the participants silently repeated the statement to themselves[5] without engaging in any conscious attempts to refute the negative self-cognitions (repeat condition). Each task block concluded with a fixation cross (rest condition) before the next block began. Prior to and following scanning, participants rated the extent to which they agreed with the presented negative self-cognition statements. Participants also completed the Perseverative

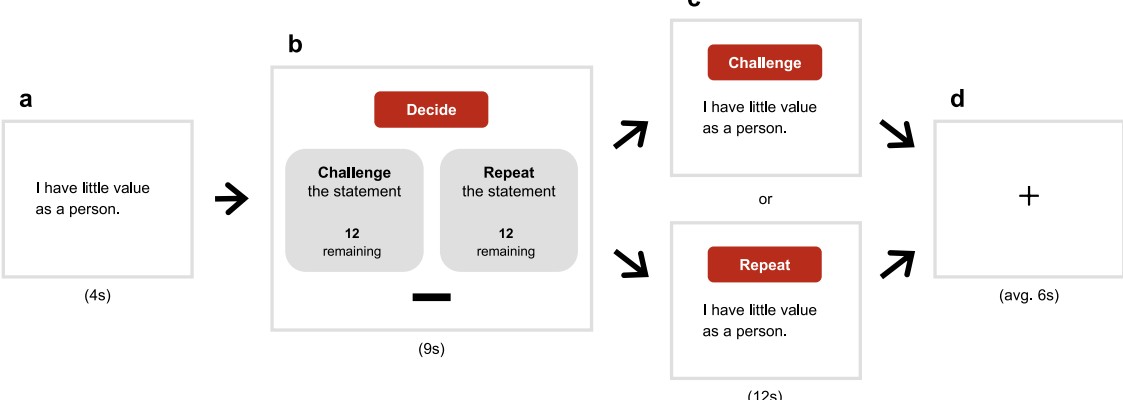

**Fig. 1 | Cognitive restructuring paradigm. a** In each of the task blocks, participants were first shown a common negative self-cognition statement on the screen (e.g., "I am incompetent in the things I do", "My value depends on my body shape") for 4 s. Each block featured a unique statement. **b** Next, participants were given 9 s to decide and select either to restructure or repeat the negative statement that was on the screen. Participants indicated their choice via an MRI-compatible button box, which moved the black cursor to the elected choice. A counter was presented under each option, indicating the remaining number of blocks they could select the respective strategy, ensuring equal numbers of statements being restructured or repeated throughout the task. **c** Once the decision period lapsed, participants were shown the same statement accompanied by an instruction to engage in their chosen strategy for 12 s (restructure = challenge condition; repeat = repeat condition). **d** A jittered fixation cross was then presented for an average of 6 s (rest condition) before the next block commenced. The replication sample group underwent an abbreviated version of this paradigm, containing 16 blocks that did not include negative self-cognition statements about food and body image.

Thinking Questionnaire[54] to assess repetitive negative thinking tendencies (reported in Supplementary Table 1).

## Habenula activity & effective connectivity during negative self-cognition processing

Mass-univariate general linear model (GLM) activation analysis revealed that the habenula had increased activity during the repeating of negative self-cognitions compared to restructuring, in addition to the right PCC, bilateral hippocampus, and the bilateral pOFC (Fig. 2, Supplementary Fig. 1 and Table 2). Of note, neural activation elicited by the repeating of negative self-cognitions was more pronounced in the right hemisphere in both the hippocampus and the pOFC compared to the left hemisphere. As illustrated in Fig. 2d, habenula response increased during both the repeating and restructuring of negative self-cognitions relative to rest. Complete GLM activation results are reported in Supplementary Table 2.

Based on these initial activation results ($n = 48$) and past neuroimaging evidence[36,38,39], a DCM network including the habenula, right PCC, right hippocampus, and right pOFC as regions-of-interest (model nodes; group-level centre coordinates reported in Table 1) was inverted for each participant to infer the modulatory influence of negative self-cognition processing on habenula effective connectivity (Fig. 2e, f)[55,56]. Our hypothesised network structure assumed the presentation of negative self-cognition statements as driving input into all regions-of-interest. To probe the effect of habenula activity on the neuronal response of other network regions, we modelled the habenula's bidirectional pathways to and from the other network nodes, in addition to their self-connections. For each of these pathways, we estimated their (1) intrinsic connectivity, which represents the context-independent interaction between the network regions, i.e., average effective connectivity across the entire paradigm; and (2) the modulatory effects of repeating or restructuring negative self-cognitions on interregional effective connectivity. Group-level effects were summarised using Parametric Empirical Bayes (PEB)[56] and thresholded at posterior probability > .95. The strength of effectivity connectivity is represented as a partial derivative, measured in hertz (Hz), that quantifies the rate at which neuronal activity in one region changes with respect to neuronal activity in another region (intrinsic connectivity) or due to an experimental input (modulatory effect).

DCM inversion ($n = 45$) and PEB revealed that both the repeating and restructuring of negative self-cognitions positively modulated the connectivity from the habenula to the PCC, such that the habenula exerted an excitatory effect on the PCC during both task conditions (Fig. 3a, b). The habenula-to-pOFC pathway was positively modulated by the restructuring of negative self-cognitions, suggesting that the habenula upregulated pOFC activity during cognitive restructuring but not the repeat condition. With regards to intrinsic effective connectivity, the PCC and the pOFC had an excitatory influence on the habenula when averaged across the entire cognitive restructuring task (Fig. 3a); whereas the habenula had an inhibitory effect on the activity of the PCC. Complete Bayesian model-averaged parameter estimates, including posterior expectation, posterior covariance and posterior probability, are reported in Table 2.

Two additional PEB models including participants' negative self-cognition endorsement and PTQ total scores as covariates did not show sufficient evidence to suggest that neither negative self-cognition endorsement nor perseverative thinking modulated habenula network dynamics (see Supplementary Table 3).

Considering the bilateral activation of the hippocampus and the OFC during the repeating of negative self-cognitions, we conducted an exploratory DCM analysis using a left-lateralised model consisting of the bilateral habenula, left PCC, left hippocampus, and the left pOFC following identical model specification and inversion procedures described above. Detailed steps for timeseries extraction and the complete Bayesian model-averaged parameter estimates are reported in the Supplementary Information. In short, the left-lateralised model revealed broadly convergent connectivity patterns as the right-lateralised model, recapitulating the excitatory connectivity from the habenula to the PCC and pOFC during cognitive restructuring (Supplementary Table 4). Additional excitatory modulatory effects were observed for the habenula-to-left hippocampus and PCC-to-habenula pathways during restructuring. The habenula-to-PCC excitatory connectivity during the repeating of self-cognitions was not observed in the left-lateralised model.

## Independent sample replication & randomised 5-fold validation

Next, we evaluated the out-of-sample validity of our effective connectivity findings of the primary right-lateralised model using an independent replication dataset ($n = 56$). Brain activation patterns in

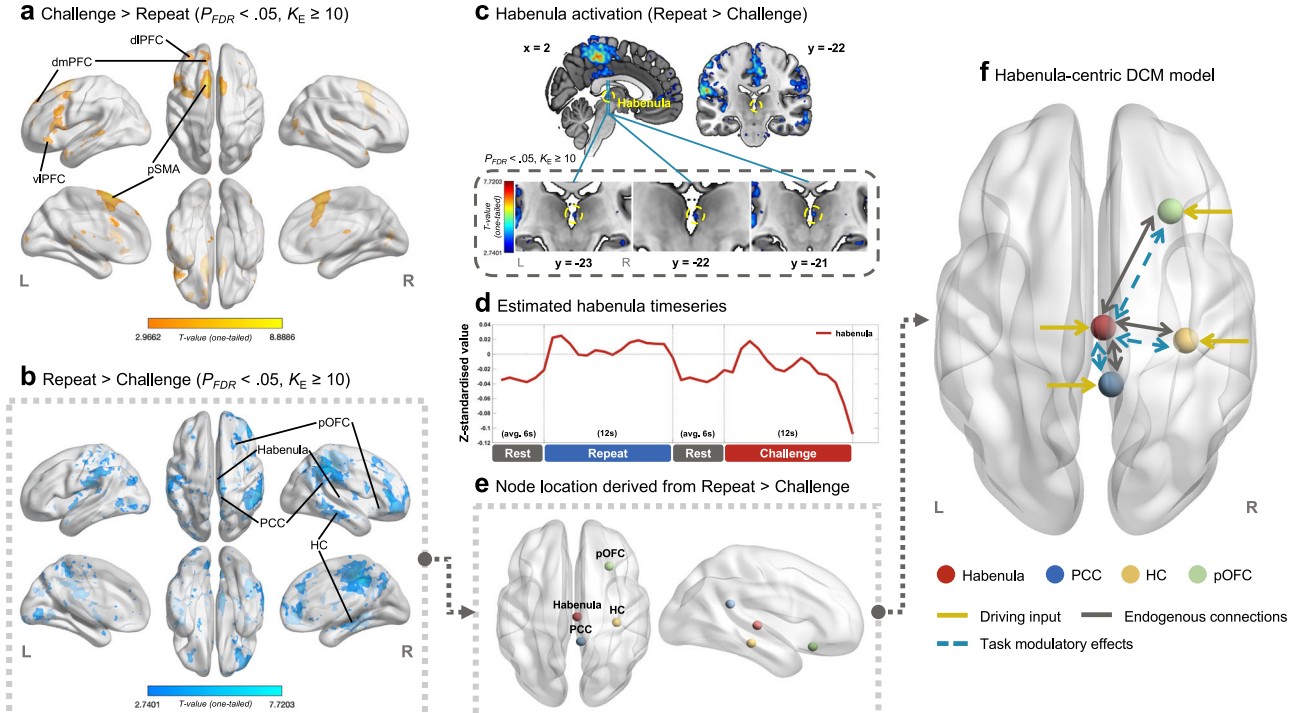

**Fig. 2 | Task-based neural activation and construction of the DCM model space.**
**a**, **b** The heatmaps display the general linear model (GLM) results of the cognitive restructuring fMRI paradigm ($P_{FDR} < 0.05$, $K_E \geq 10$), with the colour bars representing the t-statistics of the single-sample *t* test (one-tailed). The warm colour map shows brain regions with increased activity during the restructuring of negative self-cognitions compared to repeating (Challenge > Repeat). The cool colour map highlights structures showing increased activity during the repetition of negative cognitions versus restructuring (Repeat > Challenge). These results confirmed the engagement of the habenula in negative self-cognition processing and were used to inform DCM model node selection. **c** Consecutive coronal views of the neural activation results of the Repeat > Challenge contrast are presented on the MNI152 template to highlight the habenula cluster showing increased activity during the repeating of negative self-cognitions compared to restructuring. **d** The line graph plots the group-level blood-oxygen level dependent (BOLD) response (GLM-estimated) of the habenula region-of-interest across the key conditions and when averaged across task epochs (rest, repeat, rest, challenge). The habenula showed sustained activity during the repeating of negative self-cognitions (repeat condition) and evoked response during cognitive restructuring (challenge condition). **e** Based on the GLM results and past literature on habenula connectivity, the bilateral habenula, right pOFC, PCC, and the hippocampus were selected as model nodes for DCM analysis. **f** A DCM model centred on the habenula was constructed and estimated for each individual. The model assumed (1) bidirectional endogenous connections between the habenula and the other network regions (grey arrows), in addition to their self-inhibitory connectivity (not shown here); (2) driving input of the experimental stimuli into all network nodes (yellow arrows); and (3) modulatory effects of the restructuring and repetition of negative self-cognitions on the bidirectional connections between the habenula and the PCC, hippocampus, as well as the pOFC (dashed blue arrows). 3-D brain rendering were constructed in BrainNet Viewer with the MNI152 template brain[119]. *DCM* dynamic causal model, *dlPFC* dorsolateral prefrontal cortex, *dmPFC* dorsomedial prefrontal cortex, *HC* hippocampus, *L* left, *PCC* posterior cingulate cortex, *pOFC* posterior orbitofrontal cortex, *pSMA* pre-supplementary motor area, *R* right, *vlPFC* ventrolateral prefrontal cortex.

### Table 1 | Volume-of-interest (VOI) group peak coordinates in MNI space

| Region-of-interest | PCC, R | | | Hippocampus, R | | | pOFC, R | | |
|---|---|---|---|---|---|---|---|---|---|
| | x | y | z | x | y | z | x | y | z |
| Discovery sample | 6 | − 43 | 22 | 35 | − 26 | − 13 | 29 | 24 | − 16 |
| Replication sample | 9 | − 35 | 28 | 26 | − 34 | − 3 | 29 | 34 | − 14 |

*MNI* Montreal Neurological Institute, *pOFC* posterior orbitofrontal cortex, *PCC* posterior cingulate cortex, *R* right.

the replication sample during both the repeating and restructuring of negative self-cognitions were largely consistent with the discovery sample (Supplementary Figs. 2, 3 and Table 5). For the replication sample, the specification of both the DCM and group-level PEB model replicated the procedures used in the discovery model. However, we leveraged the Bayesian model-averaged group-level posterior distribution from the discovery model, specified by its mean and covariance, as an empirical prior distribution over the effective connectivity parameters of the replication PEB model[57–59]. This allowed us to inform the inversion of the replication PEB model and test the discovery model parameters in the independent dataset. The use of informed priors capitalises on the inherent advantage of the empirical Bayesian

framework to test the validity of our findings – that is, whether the same effective connectivity architecture is replicated in the independent dataset, given prior knowledge on the network dynamics derived from the discovery sample[56,60]. Here, we report the DCM results from this informed replication model and summarise the parameter estimates in Supplementary Table 6. We also include the results of the non-informed model in Supplementary Table 7.

The positive modulatory effects of the restructuring of negative self-cognitions on the habenula-to-pOFC pathway were identified again in the replication model, such that the habenula exerted an excitatory influence on the pOFC during cognitive restructuring (Fig. 4). The modulatory effects associated with the habenula-to-PCC

**a** Task-elicited habenula effective connectivity

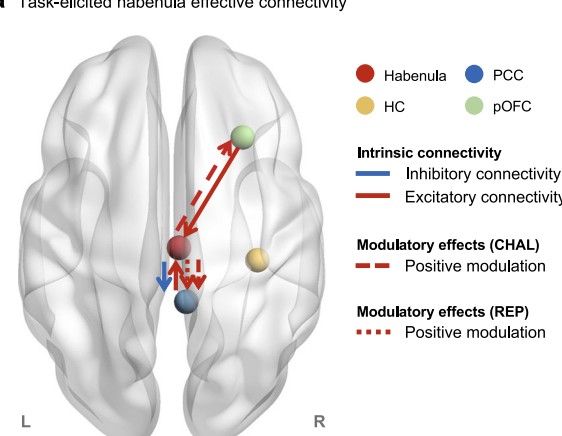

**b** Habenula efferent effective connectivity across conditions

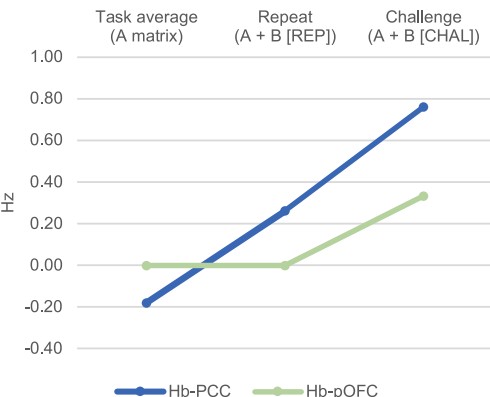

**Fig. 3 | Habenula effective connectivity in the discovery sample. a** Intrinsic connectivity and task-induced modulation of connections between the habenula and the network nodes that demonstrated strong evidence (posterior probability > .95) for a non-zero group effect are illustrated on the MNI152 template brain with BrainNet Viewer[119]. Intrinsic connectivity is represented with solid arrows while dashed arrows depict modulatory effects by each of the task conditions-of-interest – the restructuring (CHAL condition) or repeating (REP condition) of negative self-cognitions. Red arrows represent excitatory intrinsic effective connectivity or positive modulatory effects. Blue arrows show inhibitory intrinsic connectivity or negative task-induced modulation. **b** Task-induced changes in habenula effective connectivity are plotted for connectivity associated with significant modulatory effects. The first column shows the average effective connectivity throughout the task for each pathway (intrinsic connectivity; A-matrix). This represents the context-independent influence from the habenula to the PCC (blue) and the pOFC (green). The second and third columns represent the net effective connectivity of each pathway under the repeat (REP) and challenge (CHAL) conditions, respectively. This is calculated by adding or subtracting the modulatory effects from the corresponding intrinsic connectivity parameter (A-matrix + B-matrix) as the experimental input is mean-centred in our model. All connectivity estimates are in units of Hz denoting the rate of change in neural activity in the input regions (e.g., PCC, pOFC) due to neural response of the output region (i.e., habenula). Source data are provided as a Source Data file. *CHAL*, challenge condition, *Hb* habenula, *HC* hippocampus, *Hz* hertz, *L* left, *PCC* posterior cingulate cortex, *pOFC* posterior orbitofrontal cortex, *R* right, *REP* repeat condition.

pathway during restructuring and repeating were positive, though they did not surpass the posterior probability > .95 threshold.

To further assess whether these findings were influenced by individual participant variance, we conducted a randomised stratified 5-fold validation using the combined dataset (including both the discovery and replication groups) and examined the consistency of connectivity results across the validation subsamples. As reported in Supplementary Table 8, the subsamples were comparable on sex ($\chi^2 = 7.60$, $P_{bonf.-corrected} = .535$), age ($F_{4,96} = 0.69$, $P_{bonf.-corrected} = 1.000$), perseverative thinking ($F_{4,96} = 1.39$, $P_{bonf.-corrected} = 1.000$), and endorsement of negative self-cognitions ($F_{4,96} = 0.20$, $P_{bonf.-corrected} = 1.000$). Consistent with the replication model described above, the 5-fold validation group-level DCM models were furnished with the empirical prior distribution derived from the posterior distribution of the discovery model. Through this procedure, we found very strong evidence (posterior probability > .95) supporting the positive modulatory connectivity of the habenula-to-pOFC during the restructuring of negative self-cognitions in 4 out of the 5 subsamples (Supplementary Fig. 4). In addition, very strong evidence (posterior probability > .95) of positive modulation of the habenula-to-PCC connection during the restructuring and repeating conditions were replicated in 3 out of the 5 subsamples, upholding the reliability of our connectivity results despite moderate levels of inter-sample variability.

## Discussion

In this study, we combined DCM and 7T fMRI to infer the role of a habenula-centric circuitry during the processing of negative self-cognitions. In line with our hypothesis, we observed increased habenula activity during the repeating of negative self-cognitions compared to restructuring. Using data from two independently acquired samples, as well as a randomised 5-fold validation, we identified a reliable network structure revealing excitatory effective connectivity from the habenula to the pOFC during the restructuring of negative self-cognitions. In the discovery sample, we identified excitatory effective connectivity from the habenula to the PCC during both the

repeating and restructuring of negative self-cognitions. These findings provide distinct insights into the habenula's functional influence on key nodes of the default mode network and cognitive control network to support self-related higher-order cognitions in humans, thereby broadening our understanding of habenula function to encompass domains not limited to external primary reward or punishment.

Although both the restructure and repeat task conditions modulated habenula activity, it is worth noting that habenula response was heightened during the repeating of negative self-cognitions. As negative self-cognitions were not consciously restructured and reduced in intensity during the repeat condition, increased habenula activity may reflect the sustained signalling of negative valence induced by the repetition of negative self-cognition statements. This aligns with the understanding that habenula activity encodes the negative motivational value of external punishment or unrewarding outcomes to influence behavioural response[18,61], suggesting parallel neural processes between the encoding of the aversiveness of negative self-cognitions and negative reward signalling. This valence-based information may then be integrated into higher-order cognition subserved by other cortical systems[62].

Our model demonstrated that both the restructuring and repeating of negative self-cognitions positively modulated habenula-to-PCC connectivity, such that the habenula consistently exerted an excitatory influence on the PCC during engagement with negative self-cognitions. As a core node of the default mode network, the PCC has been posited to play a coordinating role in the flexible attentional switch between internal and external environments[63,64]. Through interactions with the frontoparietal executive control regions and the salience network (e.g., anterior cingulate cortex, anterior insula), the PCC receives information regarding the personal relevance of the task at-hand to inform attentional resource allocation[65,66], with sustained PCC activity facilitating more self-oriented cognition[67]. Crosstalk between the habenula and the PCC may similarly allow the incorporation of valence and motivational value to tilt attentional balance towards self-referential processes when faced with negative self-

**Table 2 | Bayesian model-averaged DCM parameters for endogenous and modulatory connections in the discovery sample**

| Connection | Ep | Cp | PP |
|---|---|---|---|
| **Endogenous connections[a] (A-matrix)** | | | |
| Habenula → Habenula | − 0.57 | 0.0016 | 1.00* |
| Habenula → PCC | − 0.18 | 0.0006 | 1.00* |
| Habenula → Hippocampus | <0.01 | <0.0001 | .00 |
| Habenula → pOFC | <0.01 | <0.0001 | .00 |
| PCC → PCC | − 0.49 | 0.0017 | 1.00* |
| PCC → Habenula | 0.07 | 0.0005 | .99* |
| Hippocampus → Hippocampus | − 0.44 | 0.0020 | 1.00* |
| Hippocampus → Habenula | − 0.09 | 0.0016 | .93 |
| pOFC → pOFC | − 0.27 | 0.0021 | 1.00* |
| pOFC → Habenula | 0.13 | 0.0011 | 1.00* |
| **Modulatory connections[b] (B-matrix)** | | | |
| **Challenge (CHAL)** | | | |
| Habenula → PCC | 0.94 | 0.0121 | 1.00* |
| Habenula → Hippocampus | <0.01 | <0.0001 | .00 |
| Habenula → pOFC | 0.33 | 0.0052 | 1.00* |
| PCC → Habenula | <0.01 | <0.0001 | .00 |
| Hippocampus → Habenula | < − 0.01 | <0.0001 | .00 |
| pOFC → Habenula | <0.01 | <0.0001 | .00 |
| **Repeat (REP)** | | | |
| Habenula → PCC | 0.44 | 0.0099 | 1.00* |
| Habenula → Hippocampus | < − 0.01 | <0.0001 | .00 |
| Habenula → pOFC | < − 0.01 | <0.0001 | .00 |
| PCC → Habenula | <0.01 | <0.0001 | .00 |
| Hippocampus → Habenula | <0.01 | <0.0001 | .00 |
| pOFC → Habenula | <0.01 | <0.0001 | .00 |

[a]Endogenous parameters reflect the average effective coupling between regions across experimental conditions (context-independent).

[b]Modulatory parameters reflect the changes in effective coupling between regions induced by cognitive reappraisal (content-dependent).

*Posterior probability (PP) exceeding .95 provides sufficient evidence for a non-zero group effect[56].

Cp posterior covariance, Ep posterior expectation, PCC posterior cingulate cortex, pOFC posterior orbitofrontal cortex, PP posterior probability.

related cognitions. This hypothesis accords with the view that connectivity between the habenula and the default mode network potentially reflects an integrative self-monitoring process, with the value of negative stimuli being signalled by the habenula[39]. Moreover, the PCC is integral to the generation of a unitary representation of the self that is then gated into conscious awareness via activity in the mPFC[63,68]. This is particularly relevant during autobiographical memory recall, where the PCC-mediated self-conceptualisation represents egocentric information associated with prior experiences[65,67]. Via the habenula-to-PCC pathway, negative valence encoded in the habenula may be attributed to mental representations of the self during associations triggered by the internal recital of negative statements or during the conscious recollection of personal experiences required to refute them.

The habenula exerted a distinct excitatory effect on the pOFC during the restructuring of negative self-cognitions – an effect that was reliably detected in both the discovery and replication samples. Restructuring negative self-cognitions not only requires the sustaining of complex self-concepts, subserved by the default mode network[68], but also the manipulation of self-representations, which has been shown to involve frontostriatal valuation and cognitive control

circuits[50,69]. Our observed widespread activation in the lateral PFC control regions and the pre-supplementary motor area extending across the dorsomedial PFC, as well as the caudate and the thalamus, during the restructuring of negative self-cognitions compared to repeating further supports this and is largely consistent with past studies on cognitive emotion regulation highlighting the interaction between frontoparietal and subcortical structures when down-regulating negative affect[62,70]. Under this framework, the PFC is thought to carry out reflective and contextual appraisals/evaluations, while subcortical and brainstem regions supply more basic information defining the affective state (e.g., salience, valence, physiological response)[71]. The observed excitatory connectivity from the habenula to the pOFC concurs with these reports and suggests a previously undescribed role of the habenula in shaping adaptive responses to negative cognitions. Besides negative valence, the habenula is known to be sensitive to trial-to-trial feedback for reward outcomes[72,73], and to contribute to the flexible modification of action strategies in reaction to aversive outcomes[14,74]. These findings point to the habenula's contribution to tracking the effectiveness of behavioural responses in relation to outcome expectations and feedback throughout changing contexts. In this regard, excitatory connectivity from the habenula to the pOFC may act as a pathway through which the expectation and outcome of the restructuring effort is transmitted from the midbrain to the prefrontal cortex.

Our findings are in accordance with the growing consensus that the OFC constructs and maintains a cognitive map defining the current task space, in which multiple sources of information relevant to decision-making (e.g., action-outcome value, emotion, memory) are synthesised[75–77]. The recruitment of the OFC is necessary in situations requiring mental simulation or future inferences, where values and predictions of possible outcomes associated with each choice option need to be computed with continuously updated information[78,79]. During the cognitive restructuring paradigm used in this study, participants were presented with a different statement on each trial and could not rely on previously formulated arguments to restructure the negative self-cognitions. Rather, participants need to dynamically adjust their cognitive strategies and to conceive new rebuttals in response to different self-cognition statements. Each restructuring strategy may represent an alternative task state that necessitates OFC-mediated representation and outcome-value computation[77]. This real-time evaluation likely incorporates the moment-by-moment feedback about the expected and actual effectiveness of the restructuring strategy encoded by the habenula[74,80], a process potentially subserved by the restructuring-induced positive modulation of the habenula-to-pOFC excitatory connectivity.

Mizumori and Baker[81] recently synthesised findings from animal models to hypothesise that the habenula integrates action-outcome valuation from the mPFC and information about the organism's internal state from the subcortex (e.g., lateral hypothalamus, entopeduncular nucleus) to signal the effectiveness of behaviours in relation to a contextual goal[81,82]. They proposed that the habenula encodes the decision to continue or alter the current course of action while simultaneously relaying this information to the hippocampus and the mPFC where subsequent actions may be updated and evaluated[81]. How this model applies to humans remains unknown. Nevertheless, our current findings support the role of habenula-frontal cortex interactions in the adaptive processing of negative cognitions and significantly expand the evidence base for the human habenula's role in complex, higher-order cognitive processes[83,84].

Contrary to our hypotheses, we did not find sufficient evidence to support modulation in connectivity between the habenula and the right hippocampus during negative self-cognition processing. However, the exploratory left-lateralised model revealed excitatory modulation from the habenula to the left hippocampus during the challenge condition, suggesting that the two hippocampi may be

**a** Task-elicited habenula effective connectivity of the Replication model

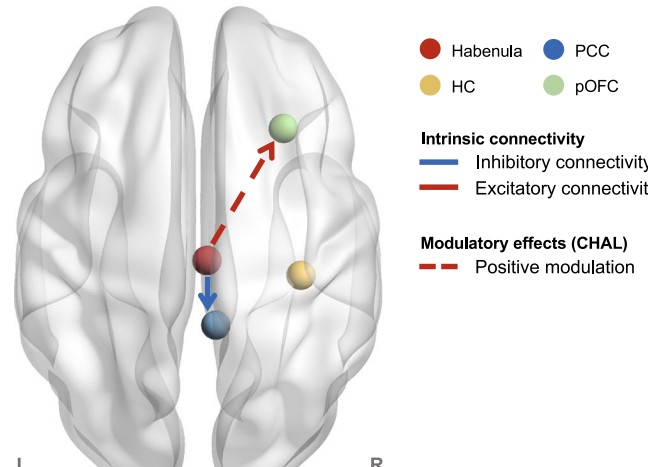

**b** Habenula connectivity across conditions

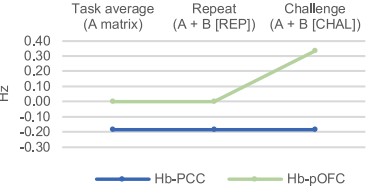

**c** Modulatory connectivity comparison

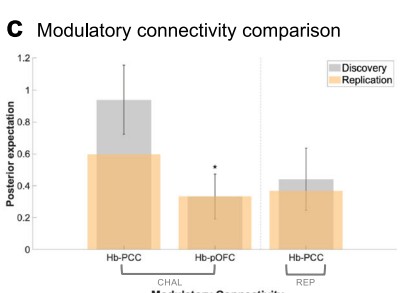

**Fig. 4 | Habenula effective connectivity in the replication sample. a** Intrinsic and task-induced modulation of habenula connectivity with strong evidence (posterior probability > .95) for a non-zero group effect are displayed on the MNI152 template brain using BrainNet Viewer[119]. Solid arrows represent intrinsic connectivity, while dashed arrows illustrate modulatory effects by the task conditions – the restructuring (CHAL condition) or repeating (REP condition) of negative self-cognitions. Red arrows show excitatory intrinsic connectivity or positive modulatory effects. Blue arrows indicate inhibitory intrinsic connectivity or negative task-induced modulation. **b** Changes in habenula effective connectivity are plotted for connectivity showing significant task-induced modulation. The first column quantifies the average effective connectivity throughout the task for each pathway (intrinsic connectivity; A-matrix), representing the context-independent influence from the habenula to the PCC (blue) and the pOFC (green). The second and third columns show the net effective connectivity of each pathway under the repeat (REP) and challenge (CHAL) conditions. As the experimental input is mean-centred in our model, this is calculated as the intrinsic connectivity parameter plus the modulatory effect of the corresponding task condition (A-matrix + B-matrix). All connectivity estimates are in units of Hz denoting the rate of change in neural activity in the input regions (e.g., PCC, pOFC) due to neural response of the output region (i.e., habenula). **c** The bar graph depicts modulatory connectivity of the discovery model (*n* = 45; grey) superimposed with the posterior expectation of the replication model (*n* = 56; orange). The bars represent the Bayesian model-averaged (BMA) connectivity strength estimates of the corresponding network connection in the models, and the whiskers show the 95% confidence interval (CI) centring the discovery model parameter estimates derived from the posterior covariance matrix (spm_plot_ci.m). An asterisk is placed above the bars for connectivity replicated across the models, which are identified based on the replication model connectivity with posterior expectations that are within the 95% CI of the discovery model estimate and surpass the posterior probability threshold (> .95). Source data are provided as a Source Data file. *CHAL* challenge condition, *Hb* habenula, *HC* hippocampus, *Hz* hertz, *L* left, *PCC* posterior cingulate cortex, *pOFC* posterior orbitofrontal cortex, *R* right, *REP* repeat condition.

differentially engaged during the restructuring of negative self-cognitions. This is broadly consistent with prior literature showing lateralised hippocampal function, such that the left hippocampus primarily encodes egocentric information and spatiotemporal association that constitute components of episodic memory, whereas the right hippocampus contributes to contextual topographic representations crucial for spatial navigation[85–87]. In early phases of learning, both hippocampi have been shown to coactivate, reflecting the novelty of a given situation and supplying complementary representations that support different aspects of memory formation[86]. The bilateral hippocampal activation observed here may reflect the effect of exposure to new self-cognition statements in each trial, while connectivity between the habenula and the left hippocampus supports episodic memory processes required for cognitive restructuring, drawing upon personal history.

We note some limitations of this work. First, the current samples included healthy participants who tended to report low levels of negative self-cognition endorsement and perseverative thinking. Thus, a floor effect may have impacted our ability to detect meaningful relationships between individual endorsement of negative self-cognitions and habenula connectivity. Relatedly, the lack of model evidence for an association between the perseverative thinking and negative self-cognition measures and habenula connectivity precluded inferences on the behavioural implication of habenula connectivity variations in healthy individuals. While the present study focused on establishing a normative role of the habenula in negative self-cognition processing, future studies would benefit from applying our habenula-centric model in populations characterised by maladaptive cognitions

(e.g., people experiencing psychopathology)[6]. It could also be important to examine whether altered habenula circuitry confers risk for mental ill-health. For example, if elevated habenula-PCC connectivity and/or dampened habenula-OFC connectivity may underpin negatively biased self-perception in depression and post-traumatic stress disorder[88,89]. Moreover, the successful inversion of DCM requires an efficient (small-scale) model structure with region selection based on a priori network hypotheses[48]. To more comprehensively characterise habenula involvement in negative self-cognitions, future studies could expand the current model by considering additional regions, such as the insula, which is functionally connected to the habenula[38], and the disruption of which has been implicated across psychiatric disorders[90]. Lastly, there were minor methodological differences between our discovery and replication datasets that may have contributed to the variance in our connectivity results. Despite these dissimilarities, we observed consistent modulation of the habenula connectivity during the restructuring of negative self-cognitions across the samples, which upholds the reliability of these findings.

Here, we present evidence demonstrating the human habenula's involvement in the higher-order processing of negative self-cognitions. We showed that habenula activity was modulated by the repeating and restructuring of negative self-cognitions. Using DCM in two independent samples, our model elucidated the directional interplay between the habenula, PCC, and the OFC, which prospectively underpins negative self-conceptualisation and the value-guided restructuring of negative self-cognitions. These findings advance our current understanding of the habenula's role in negative stimuli processing beyond primary reward and punishment to include

abstract internal experiences. A mechanistic account of habenula functioning lays the foundation to examine neural vulnerabilities contributing to maladaptive thinking patterns, and whether habenula connectivity may be predictive of response to interventions that rely on individual capacity to restructure negative self-cognitions (e.g., cognitive behavioural therapy)[91]. Our results could also inform future work assessing the habenula's potential as a treatment target to alleviate entrenched negative self-cognitions that do not respond to conventional psychotherapy alone, such as via neuro-modulatory interventions or novel pharmacological agents impacting habenula circuitry (e.g., ketamine)[92].

## Methods
### Participants
For our discovery sample, we recruited 57 healthy adults from the community via online advertisements. Inclusion criteria included: (1) being between the age of 18 and 40 years; (2) having no MRI contra-indications (e.g., pregnancy, metallic implants or claustrophobia); (3) willingness to comply with the scanning centre's healthy and safety policies (e.g., received full course of SARS-CoV-2 vaccination); (4) fluency in written and verbal English; (5) being capable of complying with study instructions. Participants were excluded if they (1) had a diagnosis of any mental disorder at the time of study participation; (2) have a past history of eating disorders, psychotic disorders, obsessive-compulsive disorder, or bipolar disorders based on The Mini International Neuropsychiatric Interview (MINI, English version 7.0.2)[93] for the Diagnostic and Statistical Manual of Mental Disorders, fifth edition (DSM-5)[94]; (3) have been diagnosed with autism spectrum disorder; (4) have major hearing or sight difficulties; or 5) have a medical or neurological condition for which they are on medication.

For our independent replication sample, we obtained MRI data from 83 healthy adults, which have also been included in previous reports[50,69,95]. Participants were eligible if they (1) aged between 18 and 40 years; (2) did not meet criteria for any mental disorders as screened using the MINI; (3) had no MRI contraindications; and (4) were proficient in English and had normal or corrected-to-normal vision.

All participants provided written informed consent and attended one testing session at the Melbourne Brain Centre Imaging Unit (The University of Melbourne, Parkville, Victoria, Australia). This study was approved by the University of Melbourne Human Research Ethics Committee (HREC 22347, 2056265).

Nine and 18 participants were initially excluded from the discovery and replication samples, respectively, due to: technical errors during MRI acquisition (discovery: 1, replication: 3), participant not completing or incorrectly completing the fMRI paradigm (discovery: 7, replication: 5), and excessive head motion (discovery: 1, replication: 10). Thus, 48 participants from the discovery and 65 participants from the replication samples were included in the GLM activation analysis. An additional 3 participants from the discovery and 9 participants from the replication samples were excluded from the DCM analysis due to a failure to extract valid time series from the regions-of-interest, resulting in 45 discovery group participants and 56 replication group participants being included in the final DCM analysis.

### Self-report measures
**Demographic information.** Age, sex (i.e., assigned sex at-birth), and ethno-cultural group based on participant self-report are summarised in Supplementary Table 1.

**Perseverative thinking questionnaire (PTQ)[54].** The PTQ is a 15-item questionnaire assessing individuals' general propensity to engage in repetitive negative thinking and its impact. Each item is rated on a 4-point Likert scale (0 = never; 4 = almost always), with higher total scores indicating stronger perseverative thinking tendencies. The PTQ

was administered to all eligible participants prior to the MRI scanning session.

**Challenging negative beliefs task questionnaire (CNBTQ).** The CNBTQ forms part of the cognitive restructuring fMRI task (see section below for detailed task description) and was administered prior to scanning and following scanning. The CNBTQ has participants rate the extent to which they endorse the negative self-cognition statements presented during the cognitive restructuring paradigm on a 7-point Likert scale (1 = strongly disagree; 7 = strongly agree). The difference in pre- and post-task ratings was used to index task-invoked shifts in negative self-cognition endorsement. Participants in the replication sample were only asked to provide post-task ratings on the statements that were restructured during the task, whereas participants in the discovery sample provided post-task ratings to all statements.

### Behavioural analyses
Demographic and behavioural variables were analysed using SPSS v.27 (IBMCorp., Armonk, NY). Changes in negative self-cognition endorsement were compared within-group using single-sample t-tests (2-tailed), and between-groups using independent sample t-tests (2-tailed), with Cohen's $d$ to quantify effect size. Likewise, two-tailed independent sample t-tests were used to assess differences in demographics and behavioural variables across the samples, whereas two-tailed Mann-Whitney U tests with rank-biserial correlation ($r$) for effect size were used when there was evidence for heteroscedasticity. One-way analysis of variance (ANOVA), with eta-squared ($\eta^2$) for effect size, was used when three or more subgroups were compared. Two-tailed Pearson's chi-squared tests were adopted to evaluate categorical variables between the groups, using $\varphi$ to indicate effect size where applicable. Bonferroni correction was applied to account for multiple comparisons.

### Cognitive restructuring paradigm
As described in Steward et al.[50], prior to scanning, participants received training to restructure negative cognitions using Socratic questioning techniques that mimic those practised in cognitive psychotherapy, such as recalling personal experiences that refute negatively biased self-conceptions[52]. Once research staff verified that the participant could correctly carry out Socratic questioning with the goal to restructure negative self-cognitions, participants completed the paradigm during MRI scanning. The cognitive restructuring task comprised one run of 24 blocks (Fig. 1). In each block, the participants were first shown a statement on screen of common negative self-cognitions about the self, food, and body image reported in the cognitive behavioural therapy and psychopathology literature (e.g., "I am incompetent in the things I do", "My value depends on my body shape"; Supplementary Table 9)[53,96]. Next, participants decided via a button press whether they would repeat or restructure the presented statement using pre-trained strategies. Participants were instructed to restructure and repeat an equal number of statements. After 9 s, the participants were shown the same statement and instructed to engage in their chosen strategy for 12 s (restructure = challenge condition; repeat = repeat condition). A fixation cross was then presented for 6 s (rest condition) before the next block began with a new statement. The cognitive restructuring paradigm used in the discovery and replication samples were identical, except that the replication group completed an abbreviated version containing one run of 16 blocks that did not include negative self-cognition statements about food and body image. All versions of the cognitive restructuring paradigm were developed and presented using E-Prime 3.0 software (Psychology Software Tools, Pittsburgh, PA).

## fMRI image acquisition

Imaging for the discovery and replication datasets was conducted on a 7-Tesla research scanner (Siemens Healthcare, Erlangen, Germany) equipped with an 8Tx/32Rx and 1Tx/32Rx head coil, respectively (Nova Medical Inc., Wilmington, MA, USA). The functional sequence was consistent across the two samples and consisted of a multi-band (factor = 6) and GRAPPA (R = 2) accelerated GE-EPI sequence[97] in the steady state (TR = 800 ms; TE = 22.2 ms; pulse angle = 45°; field of view = 20.8 cm; acquisition matrix = 130 × 130-pixel; slice thickness = 1.6 mm, no gap). Eighty-four interleaved axial slices were acquired along the anterior-posterior commissure line. In total, 946 and 628 whole-brain EPI volumes were acquired in a single run for the discovery and replication datasets, respectively, corresponding to approximately 12.6 and 8.4 min for the two task versions. High-resolution T1-weighted anatomical images were acquired for functional time-series co-registration and individualised habenula segmentation (detailed in section *Habenula segmentation & region-of-interest validation*) using the first echo of a multi-echo Magnetization Prepared 2 Rapid Acquisition Gradient Echoes sequence (ME-MP2RAGE)[98] for the discovery dataset (224 interleaved axial slices; TR = 4500 ms; TE = 2.21/4.21/6.15/8.14 ms; inversion time = 700/2700 ms; flip angle = 6/7°; field of view = 24 cm; acquisition matrix = 320 × 320-pixel; slice thickness = 0.75 mm, no gap), and a single-echo MP2RAGE[99] sequence for the replication dataset (224 interleaved sagittal slices; TR = 5000 ms; TE = 2.04 ms; inversion time = 700/2700 ms; flip angle = 4/5°; field of view = 24 cm; acquisition matrix = 320 × 320-pixel; slice thickness = 0.75 mm, no gap). Standard foam pads were used for all participants to minimise head movement during scanning. Respiratory and cardiac recordings were sampled at 50 Hz using a respiratory belt and pulse-oximeter. Information derived from these recordings were used for physiological noise correction during image pre-processing.

## fMRI image pre-processing

Imaging data was pre-processed with Statistical Parametric Mapping 12 (SPM12, v7771; Wellcome Trust Centre for Neuroimaging, London) within the MATLAB 2023a environment (The MathWorks Inc., Natick, MA). Each participant's functional time series was realigned to the mean image to correct for movement during the scan, and all images were resampled using 4th Degree B-Spline interpolation. Individual head motion was assessed with motion fingerprint[100]. Participants were excluded if they had a mean total displacement over 1.6 mm (one isotropic voxel size) and/or a maximum scan-to-scan displacement exceeding 2 mm. We imposed a stringent censoring criterion to minimise the effect of motion on signal distortion from the habenula. Each participant's anatomical T1 image was co-registered to their mean functional image, segmented and normalised to the International Consortium for Brain Mapping (ICBM) European brain template using the unified segmentation approach plus Diffeomorphic Anatomical Registration Through Exponentiated Lie Algebra (DARTEL)[101]. Lastly, the functional images were spatially normalised with the DARTEL flow fields and smoothed with a 2 mm full width at half maximum (FWHM) Gaussian kernel to preserve spatial specificity.

Cardiac and respiratory recordings were modelled using the PhysIO toolbox[102] to account for physiological noise[103]. Specifically, the Retrospective Image-based Correction function (RETROICOR)[104], respiratory response function (RRF)[105] and the cardiac response function (CRF)[106] were incorporated to correct for both periodic and variable effects of heartbeat and breathing, as well as their interaction, on BOLD signal. Anatomical component correction (aCompCor)[107] was applied to account for non-neural signal, whereby the mean and first principal components of the time series originating from the white matter (WM) and cerebrospinal fluid (CSF) were extracted using individualised DARTEL tissue maps.

## Habenula segmentation & region-of-interest (ROI) validation

Individual-specific habenula mask was generated in the native space using each participant's high-resolution anatomical image (0.75 mm isotropic) via a validated, fully automated segmentation algorithm (Multiple Automatically Generated Templates Brain Segmentation Algorithm; MAGeTbrain)[108,109]. A schematic of our image preparation pipeline is included in Supplementary Fig. 5. Using participant anatomical images and a set of habenula atlases as input, MAGeTbrain first propagates the segmentation of the atlas images to each subject image via a multi-stage image registration procedure to yield a large number of candidate segmentations for each individual (5 atlases × 21 templates = 105). Next, the candidate segmentations are fused via majority vote at each voxel, whereby the most frequently occurring label (habenula vs. non-habenula) was adopted in the final output segmentation. This approach reduces potential biases related to atlas or rater inconsistency, allows for neuroanatomical variability[43], and has been shown to produce reliable habenula volume estimates across diverse populations and image acquisition parameters[109].

The MAGeTbrain algorithm produced a binary anatomical-resolution habenula mask in the native space, which we then used to generate functional-resolution habenula ROIs in the standard space for fMRI analyses. We employed an iterative volume optimisation strategy adapted from Ely et al.[36,39] to minimise the impact of interpolation during image down-sampling, as well as the increased susceptibility to non-neural noise at a reduced resolution, to produce the most feasibly precise representation of each individual's habenula ROI in the functional space (Fig. 5):

1. Each anatomical habenula mask produced by MAGeTbrain, as well as their corresponding DARTEL segmented CSF mask, were resampled with trilinear interpolation to the functional resolution (2 mm isotropic) based on the mean functional image (Fig. 5a), normalised to the standard space using their corresponding DARTEL flowfields, and binarised at a conventional threshold of 0.2 (Fig. 5c), which resulted in habenula and CSF masks showing good spatial consistency with the original anatomical masks.
2. To maximally reduce signal contamination from the CSF, voxels in the resampled habenula mask that overlap with the resampled CSF mask (binarising threshold = 0.2) were removed.
3. The volume of the adjusted habenula mask from Step 2 was compared to the reference habenula segmentation image (i.e., subject-specific normalised habenula mask in anatomical resolution; Fig. 5b).
4a. If the volume of the adjusted habenula mask from Step 2 fell within ± 10% of the reference image, it was accepted for use as an ROI in subsequent analyses. The volume criterion was designed to optimally approximate our functional habenula ROI to the high-resolution segmentation, while taking into consideration individual anatomical variability.
4b. If the volume was above ± 10% that of the reference image, it was rejected and Step 1 was repeated with the binarising threshold adjusted upwards or downwards by 0.01, producing a slightly smaller or larger habenula mask. Steps 2-3 were then repeated. This process was iterated until the habenula mask at functional resolution was accepted (Step 4a).

This iterative volume optimisation generated individualised habenula masks with excellent spatial specificity confirmed via visual inspection and volume estimates comparable with previous reports (i.e., 30-60 mm³ combined; discovery sample: Mean = 60.93 mm³, 95% CI = [57.72, 64.13]; replication sample: Mean = 48.40 mm³, 95% CI = [46.35, 50.44])[26,109–111].

We further validated the functional specificity of our subject-specific habenula ROI via a resting-state seed-to-whole brain functional connectivity analysis conducted in CONN22a[112]. Please see Supplementary Methods for detailed information on image acquisition, pre-processing, and analysis. Using the individualised habenula ROI as a seed, we replicated the connectivity patterns reported in past studies at standard and ultra-high field strengths (Fig. 5d)[36,38,39], highlighting habenula connectivity with structures such as the ventral tegmental

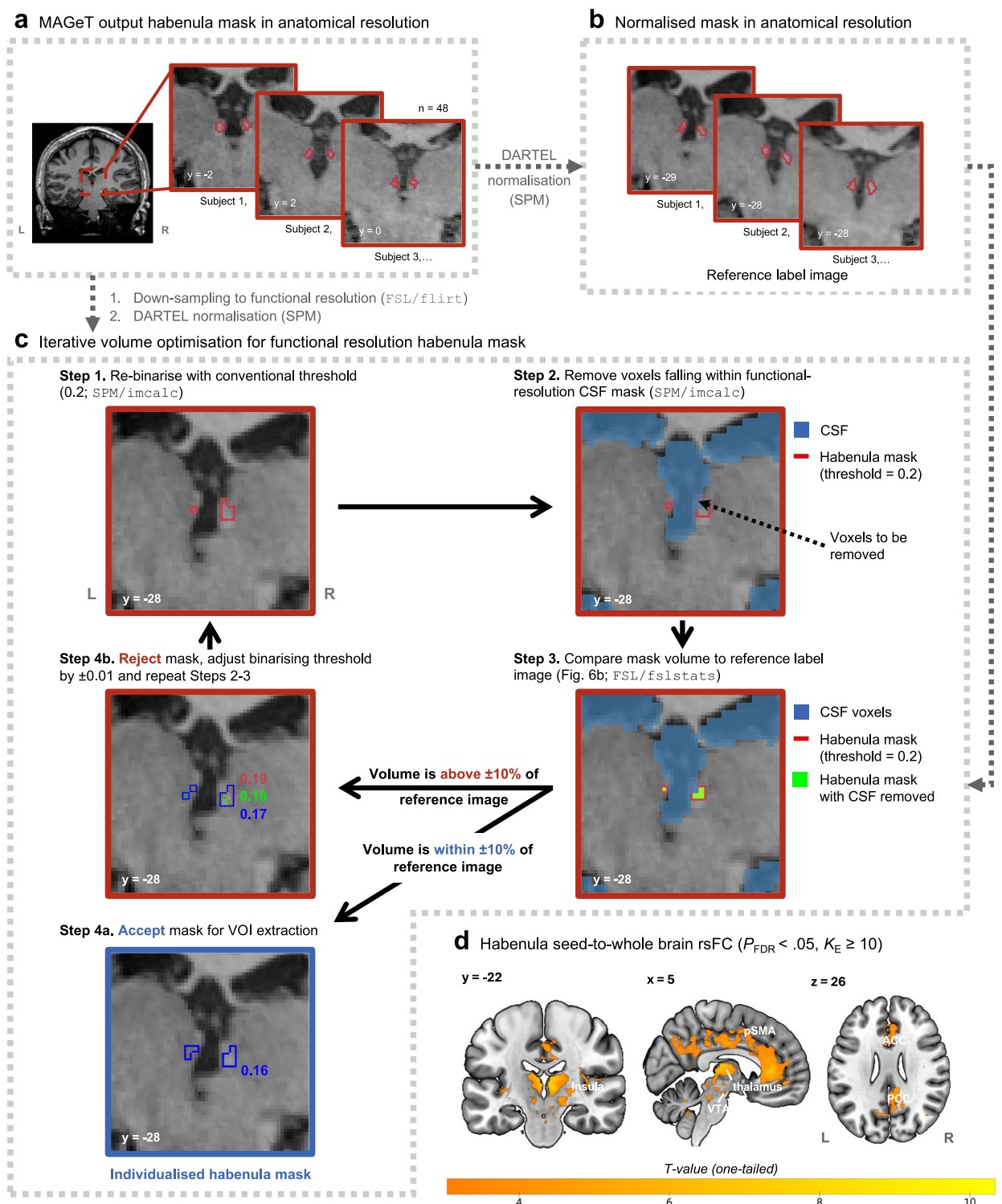

**Fig. 5 | Generation and evaluation of individual habenula ROIs. a** The MAGeT-brain algorithm produced high-resolution individualised habenula masks (red outline) using the 7-Tesla anatomical images (0.75 mm isotropic). Three example habenula masks are presented here and displayed on their corresponding T1-weighted whole-brain image in the native space. **b** The individualised habenula masks were normalised to the MNI space using the DARTEL flow fields produced during anatomical image pre-processing. These high-resolution normalised masks were used as reference label images with which we evaluated the functional resolution habenula masks. **c** To create the habenula ROIs for functional analysis, each individual's anatomical-resolution habenula mask were resampled to the functional resolution (2 mm isotropic) and transformed to the standard space. An iterative volume optimisation procedure was developed to minimise the impact of down-sampling on the habenula mask. In brief, adjustment was made to the re-binarising threshold in each iteration to ensure that the resultant habenula ROI had a volume that is within 10% of the reference label image (i.e., normalised habenula mask in anatomical resolution) after the removal of CSF voxels. **d** The subject-specific habenula ROIs in functional resolution were further validated via an rsFC analysis. Results of this analysis replicated the habenula functional connectivity pattern reported in past studies. *ACC* anterior cingulate cortex, *CSF* cerebrospinal fluid, *DARTEL* Diffeomorphic Anatomical Registration Through Exponentiated Lie Algebra, *FSL* FMRIB Software Library, *L* left, *MAGeT* Multiple Automatically Generated Templates brain segmentation, *PCC* posterior cingulate cortex, *pSMA* pre-supplementary motor area, *R* right, *rsFC* resting-state functional connectivity, *SPM* Statistical Parametric Mapping, *VOI* volume-of-interest, *VTA* ventral tegmental area.

area, thalamus, insula, anterior through to the posterior cingulate cortex, and the supplementary motor area.

## General linear modelling (GLM) analysis

Single-subject (first-level) contrast images were estimated for Challenge > Repeat and Repeat > Challenge (i.e., the entire 12 s when participants either restructured or repeated the negative self-cognition statements) to characterise changes in brain activation associated with the restructuring and repeating of negative self-cognitions, respectively. Each participant's pre-processed timeseries, 6 realignment parameters for motion artifacts, and 24 physiological noise regressors (6 cardiac, 8 respiratory, 4 cardiac × respiratory regressors, 1 RRF, 1 CRF, top principal components and mean time series of the CSF and WM) were included in the GLM analysis, with the onset times for each condition event specified and convolved with the SPM canonical hemodynamic response function (HRF). Low-frequency fluctuation was high-pass filtered at 128 Hz. The FAST method was used to estimate temporal autocorrelation resulting from our sub-second TR[113]. The first-level contrast images were brought forward to a group-level GLM (one-sample t-test, one-tailed). All GLM analyses were thresholded at whole brain, false discovery rate (FDR) corrected $P_{FDR} < 0.05$, $K_E \geq 10$ voxels.

## Dynamic causal modelling (DCM)

DCM is a Bayesian framework used to infer the directional and causal influence that brain regions exert on one another (i.e., effective connectivity)[48]. DCM enables inferences to be made by simulating neuroimaging timeseries via a generative model that is grounded in empirical knowledge of neuronal processes and how they translate to observable responses (e.g., BOLD signal)[47,114]. Within a set of researcher-specified hypotheses regarding network structure, DCM estimates connectivity parameters through model inversion in a manner that optimises the trade-off between model fit to the observed data and model complexity[55]. This balance is reflected by free energy, which approximates the (log) model evidence used for model comparison and hypothesis testing between competing models[115]. The model with the most positive free energy reflects a parsimonious and physiologically plausible account of the dynamic interactions between unobserved neuronal populations[116].

**Model space & timeseries extraction.** Our DCM network included the bilateral habenula as a single ROI, as well as brain regions that were co-activated during Repeat > Challenge that have structural/functional connectivity with the habenula based on prior literature[24,36,38]. Specifically, the right PCC, hippocampus, and pOFC were included due to their central role in self-referential thinking, episodic memory and learning, as well as adaptive processes that are engaged during negative self-cognition processing. Representative time series (volume-of-interest; VOI) were extracted from these areas for each of the subjects following published guidelines[55].

The habenula ROI was delineated with the individualised mask described above. Whereas all other network region ROIs were defined as a 4 mm radius sphere centred around the individual neural activation maxima under the Repeat > Challenge contrast, constrained to be within 8 mm from each dataset's group peak (Table 1). Volume-of-interest from each of these regions were calculated using SPM as the principal eigenvariate of all voxels within the respective ROI that showed meaningful activation for the task contrast ($p < .05$, uncorrected) at the single subject level. In the case when an ROI contains no voxel surpassing the preset threshold, the statistical threshold incrementally relaxed to $p < .5$ (uncorrected) until a peak coordinate can be identified. This approach ensured minimal exclusion of participant data from our analyses as subjects lacking strong responses in a brain region or experimental condition may nevertheless provide useful information about other regions, conditions, and individual

variability[55]. Of note, the contrast image used for VOI extraction was denoised with the motion and physiological nuisance regressors described in the General Linear Model section above, except the aCompCor components. This adjustment took into consideration the explicit removal of the CSF voxels from our individualised habenula mask and the habenula's high WM density to prevent overcorrection of the extracted timeseries, while minimising non-neural noise. Individual timeseries were additionally pre-whitened to mitigate serial correlations, high-pass filtered, and nuisance effects not covered by the Effects of Interest F-contrast were regressed out of the timeseries (i.e., 'adjusted' to the F-contrast). Using this procedure, we extracted a complete set of VOIs for 45 and 56 individuals from the discovery and replication cohorts, respectively, which were then included in the connectivity analyses.

**Model specification & estimation.** Our model was specified with SPM DCM 12.5 and featured (1) the endogenous connections between and within each target region (A-matrix), (2) the modulatory effect of the task conditions on inter-region connectivity (B-matrix), as well as (3) the driving influence of the task stimuli (C-matrix). All experimental input was mean-centred to aid parameter interpretability. As illustrated in Fig. 2f, the full model assumed bidirectional endogenous connections of the habenula to and from the other regions in addition to their self-inhibitory connectivity, modelling the average connectivity parameters throughout the experiment that is independent of condition effects. We allowed the challenge and repeat conditions to modulate the connectivity between the habenula and other network nodes to characterise changes to habenula connectivity during negative self-cognition processing. Lastly, the aggregate of the 4 s of statement presentation and 12 s of restructuring/repeating negative self-cognitions was specified as the driving input to all nodes, reflecting the engagement of these regions during the exposure to negative self-cognition statements. This full model was estimated and evaluated for each subject, yielding individual posterior connectivity parameter estimates and their posterior probability.

**Parametric empirical Bayes.** Next, a group-level summary of the connectivity parameters was obtained via Parametric empirical Bayes (PEB)[56]. PEB is a hierarchical model that incorporates both the subject-level parameter estimates and their uncertainty (i.e., posterior covariance) to the group level. Contrary to the standard summary statistics approach, the PEB framework effectively downweights data with noise and uncertain individual estimates to produce more reliable population connectivity estimates[56,60]. In addition, each level of the PEB hierarchy serves as a prior on the estimates of the level below it. This can improve the precision of individual parameter estimates by incorporating knowledge around task effects garnered from the cohort.

Our PEB model was designed to investigate the between-subject commonalities in connectivity parameters for each of the samples. The design matrix included an intercept term (single column of ones) denoting the overall mean connectivity[56]. Parameters from both A- and B-matrices were summarised in this model to account for potential conditional dependency. A generic prior distribution were adopted for the discovery model with no assumptions around the strength and variance of network connectivity[55]. Once this full PEB model was inverted, Bayesian Model Reduction (BMR) was used to search and compare the relative evidence of possible reduced models, iteratively pruning parameters that do not contribute to an increase in model evidence[117,118]. Bayesian Model Averaging (BMA) was then used to aggregate the parameters of the reduced models, weighted by the corresponding model's posterior probability, to provide the final group summary[56]. The BMA parameters were thresholded at posterior probability > .95, indicating sufficient evidence for a non-zero group effect.

The two PEB models investigating the effect of negative self-cognition endorsement and repetitive negative thinking tendencies on habenula connectivity included the individual pre-task CNBTQ and PTQ total scores, respectively, as covariate regressors, in addition to the intercept term. As all regressors were mean-centred, the between-subject effects could be quantified as the addition to or subtraction from the overall mean connectivity estimates[56].

For the replication and 5-fold validation models, the generic prior distribution was substituted in the PEB model with the posterior distribution derived from the discovery model BMA summary. The discovery model posteriors served as 3rd-level empirical priors to constrain the 2nd-level (group-level) estimates of the replication and 5-fold validation models. As a result of BMR applied during the discovery model estimation, only connectivity with a non-zero posterior probability were evaluated in the replication and validation PEB models. This approach allows the testing of the discovery model parameters on the independent dataset and effectively incorporates empirically derived beliefs around network dynamics and their degree of uncertainty to refine the estimation of connectivity parameters in new populations[56,60].

### Reporting summary
Further information on research design is available in the Nature Portfolio Reporting Summary linked to this article.

## Data availability
Deidentified effective connectivity data for this study are publicly available at https://github.com/pohankung/NegativeBeliefs_Habenula_DCM and deposited in Zenodo under accession code https://doi.org/10.5281/zenodo.15151291. Source data are provided in this paper.

## Code availability
Scripts used to generate the main results and figures of this study are available at https://github.com/pohankung/NegativeBeliefs_Habenula_DCM and deposited in Zenodo under accession code https://doi.org/10.5281/zenodo.15151291. Custom code was written in MATLAB 2023a. Statistical Parametric Mapping 12 (SPM12) and FMRIB Software Library (FSL) 6.0.6.5 were used for MRI processing.

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

## Acknowledgements

We thank James Agathos, Carly Beveridge, Lieselotte Claes, Yingliang Dai, Elizabeth Haris, Sevil Ince, Amy Nielson, Mia O'Shea, Tudor Sava, Braden Thai, and Andong Zhou for their contribution to data collection. We acknowledge the technical and scientific assistance of the Australian National Imaging Facility – a National Collaborative Research Infrastructure Strategy (NCRIS) capability at the Melbourne Brain Centre Imaging Unit (MBCIU), The University of Melbourne. The multiband fMRI sequence was generously supported by a research collaboration agreement with CMRR, the University of Minnesota. Siemens Healthineers (Germany) provided the MP2RAGE sequence. This study was supported by the National Health and Medical Research Council of Australia (NHMRC)/Medical Research Future Fund (MRFF) Investigator Grant (MRF1193736), a Brain & Behaviour Research Foundation (BBRF) Young Investigator Grant and a University of Melbourne McKenzie Fellowship to T.S. Collection of the replication dataset was supported by an NHMRC Project Grant (1161897) to B.J.H. and an NHMRC Program Grant (1073041) to K.L.F.

## Author contributions

P.-H.K., B.J.H., and T.S. conceived the general concept of this study, designed the experiment, and developed the model with input from M.D.G. and E.G.-H. P.-H.K., E.G.-H., B.J.H., K.L.F., H.C., P.S., R.M.B., B.A.M., R.K.G., and T.S. aided with data collection and the crafting of the imaging protocol. P.-H.K. and T.S. conducted data analysis with support from M.D.G. and A.J.J. B.J.H., C.G.D., K.L.F., P.S., R.M.B., and T.S. provided supervision throughout the study. P.-H.K. and T.S. wrote the original draft of the manuscript. All authors reviewed and approved the final edit of this manuscript.

## Competing interests

The authors declare no competing interests.
