## [Transparent Peer Review file · Nature Communications]

Habenula contributions to negative self-cognitions

Corresponding Author: Dr Trevor Steward

Version 0:

Reviewer comments:

Reviewer #1

(Remarks to the Author)

I congratulate the authors for presenting one of the most rigorous and methodologically well-executed studies of human habenula function to date. I am particularly impressed by the attention given to maintaining high spatial fidelity throughout the study, and find the approaches to fMRI acquisition, Hb ROI definition, task design, preprocessing/denoising, and statistical modeling all highly satisfactory. The inclusion of an independent verification sample is also a prominent strength. My only concerns relate to the selection of additional ROIs for the DCM analysis, and I am confident the authors can address these issues on revision.

1. The decision to focus on the PCC, hippocampus, and pOFC for DCM was a key study design consideration and needs to be more fully justified. The insula comes to mind as another region that has a known functional relationship with the Hb, is important in processing of negatively valenced information, and appears to be activated by the authors' Repeat > Challenge task contrast. The authors don't necessarily need to expand their set of DCM ROIs, but they should make clearer their rationale for selecting only these ROIs and rejecting other candidate ROIs.

2. Except for processes known to be strongly lateralized (e.g. language processing), I tend to be skeptical of asymmetric brain models/interpretations. I appreciated that the authors used a bilateral Hb ROI for functional analyses, and I understand that the selection of right PCC/hippocampus/OFC ROIs was driven by the stronger activation observed in these regions during the task, but the supplementary figures appear to show at least some left-hemisphere activation in these regions as well. The authors should repeat their DCM analysis including both left and right hemisphere ROIs to clarify whether the effects described in this paper are truly restricted to the right hemisphere, are weaker but still apparent in the left hemisphere, or perhaps whether some more complex interaction between hemispheres is at work. Given the procedure for defining subject-level activation peaks, this will likely reduce the overall power relative to the right-hemisphere-only version, so the authors may deem it more appropriate to present this as a supplementary analysis/result. If such an analysis is not feasible for this dataset, they should state clearly why this is the case.

Reviewer #2

(Remarks to the Author)

Thank you for the opportunity to review the manuscript titled "Habenula neural circuitry drives negative self-cognitions." This study is a novel and significant contribution, leveraging 7T fMRI and dynamic causal modeling (DCM) to investigate the role of the habenula in processing and restructuring negative self-cognitions. The work makes meaningful advancements in understanding the habenula's involvement in higher-order cognitive processes and its interactions with key brain regions such as the posterior cingulate cortex (PCC) and posterior orbitofrontal cortex (pOFC).

While the study is innovative and well-executed, there are areas for further clarification and improvement, particularly concerning the title, the depth of behavioral analysis, and the broader clinical implications. Below, I outline my detailed evaluation:

1. Title: Potential Overstatement

The title suggests a deterministic role of the habenula in driving negative self-cognitions. However, the findings primarily indicate that the habenula modulates connectivity with the PCC and pOFC during specific cognitive processes. The causal mechanisms underlying negative self-cognition remain to be fully established.

2. Limited Connection to Clinical Pathology

While the study provides valuable insights into habenula-centric networks in healthy individuals, it lacks direct links to pathological conditions like depression, where negative self-cognition is prominent. The authors could further discuss how these findings might be extended to clinical populations.

For example, could resting-state functional connectivity or spectral DCM in patients with depression validate the habenula's role in pathological negative cognition? If the discovery sample includes resting-state fMRI, it would also be interesting to test whether the spectral DCM of the habenula and PCC shows a pattern consistent with the task-based findings, which would strengthen the study's conclusions.

3. Behavioral Data Analysis

The manuscript could delve deeper into the behavioral data from the cognitive restructuring task, particularly decision-making during the "decide phase" (restructure vs. repeat). For example:

Are there individual differences in decision patterns, and how might they correlate with habenula connectivity?

Could modeling approaches, such as drift-diffusion models (DDM), provide insights into decision biases or latencies linked to negative cognition?

By exploring these aspects, the connection between neural findings and their cognitive-behavioral implications would be significantly strengthened.

4. Discussion of Limitations

The discussion section acknowledges the healthy sample as a limitation but could expand on how this affects the study's generalizability to clinical contexts. For example, it would be interesting to test whether a clinical sample shows an atypical habenula-based pattern that drives the pathology of negative cognition.

Additionally, the lack of significant habenula-hippocampus connectivity findings deserves more attention. Is this due to task design limitations, sample characteristics, or other factors?

5. Clinical Implications

While the habenula's role in negative self-cognition is intriguing, the practical applications of these findings remain underexplored.

It would be interesting to highlight the potential for neuromodulatory interventions targeting the habenula to influence cognitive restructuring or alleviate maladaptive negative self-cognitions. The authors could also discuss the relevance of these findings for optimizing therapeutic techniques, such as CBT, by integrating neural insights.

Reviewer #3

(Remarks to the Author)

This study exemplifies excellence in neuroimaging research, demonstrating a high level of methodological rigour and innovation. The use of high-magnetic field strength provides exceptional data quality. At the same time, the advanced application of Dynamic Causal Modeling (DCM) analyses showcases the authors' expertise in utilizing cutting-edge analytical tools. The inclusion of subcortical regions, often overlooked in similar studies, further enhances the depth and comprehensiveness of the work. Additionally, the innovative task design reflects a thoughtful approach to addressing a timely and significant research question. Including two well-powered datasets ensures the reliability and robustness of the findings, making the results highly impactful and generalizable.

The manuscript is very well written, with a clear and coherent structure. The figures are of excellent quality and effectively support the interpretation of the results. Despite the complexity of the analyses, the authors provide sufficient clarity for the study to be accessible to readers with varying levels of expertise. This work represents a significant advance in the field of neuroimaging and aligns perfectly with the high standards of this journal.

In summary, this is a well-executed and timely contribution to the field of neuroimaging research. I strongly recommend this manuscript for publication, as it meets the journal's high standards and advances the state of the art in multiple dimensions.

While the manuscript is already of exceptional quality, I offer the following minor suggestions to enhance its clarity and impact further:

1. If applicable, including a table within the main text with the coordinates of the ROIs used in the DCM analysis would be beneficial. While these coordinates are reported in Table S4, it is unclear what specific regions were used as ROIs in that table.
2. The discussion of the task-based GLM results, reported in the supplementary material on page 3, could be moved to the main results section as this analysis forms the foundation for the subsequent DCM analysis. Additionally, these findings overlap significantly with activity related to reappraisal, which could be discussed further.
3. Figure 2: The labels in the legend of Fig. 2A, 2B, 2C, and 2D are tiny—please increase the font size. The coordinates in Fig. 2C are challenging to read, and highlighting the habenula activity with a circle could enhance clarity.
4. Figures 2F, 3A, and 4A: The color of the pOFC in the brain images does not match the legend. Please ensure that the colours are consistent throughout.
5. Figure 4B: The abbreviation "CHAL" should be explained in the figure legend for clarity.

Reviewer #4

(Remarks to the Author)

Version 1:

Reviewer comments:

Reviewer #1

(Remarks to the Author)

I am satisfied with the changes made in response to my initial review and have no further critiques.

Reviewer #2

(Remarks to the Author)

The authors have addressed my concerns. Good work!

Reviewer #3

(Remarks to the Author)

Reviewer #4

(Remarks to the Author)

Response to reviewers

Manuscript #: NCOMMS-24-82418

Submission title: Habenula contributions to negative self-cognitions

We would like to thank the reviewers for their positive appraisal of our manuscript, as well as their recognition of the novelty, rigour, and potential impact of our study. We appreciate their constructive feedback, which was very helpful in improving the manuscript. We have provided point-by-point responses to all reviewer comments below.

Reviewer #1 (Remarks to the Author):

I congratulate the authors for presenting one of the most rigorous and methodologically well-executed studies of human habenula function to date. I am particularly impressed by the attention given to maintaining high spatial fidelity throughout the study, and find the approaches to fMRI acquisition, Hb ROI definition, task design, preprocessing/denoising, and statistical modeling all highly satisfactory. The inclusion of an independent verification sample is also a prominent strength. My only concerns relate to the selection of additional ROIs for the DCM analysis, and I am confident the authors can address these issues on revision.

1. The decision to focus on the PCC, hippocampus, and pOFC for DCM was a key study design consideration and needs to be more fully justified. The insula comes to mind as another region that has a known functional relationship with the Hb, is important in processing of negatively valenced information, and appears to be activated by the authors' Repeat > Challenge task contrast. The authors don't necessarily need to expand their set of DCM ROIs, but they should make clearer their rationale for selecting only these ROIs and rejecting other candidate ROIs.

Reply: Thank you for the insightful comment. In specifying a habenula-centric network model of negative self-cognitions, we prioritised key brain regions implicated in adaptive cognitive control (i.e., pOFC), self-referential cognitions (i.e., PCC), and autobiographical memory (i.e., hippocampus), and that have also been shown to be functionally related to the habenula (in terms of resting-state functional connectivity). We have provided further justification for our region selection in the Introduction section as quoted below. Our neural activation results also align with past literature to show that the PCC, pOFC, and the hippocampus prominently coactivate with the habenula during the repeating of negative self-cognitions and were therefore included as ROIs of the DCM analysis.

Lines 85-96: *"In addition to reward processing regions, resting-state imaging studies have found that habenula activity is correlated with activity of the orbitofrontal cortex (OFC), posterior cingulate cortex (PCC), and the hippocampus – key regions in networks implicated in outcome valuation, self-referential cognition, and memory functioning^{36, 38, 39}. The OFC has been hypothesised to receive valence-based information from the habenula to compute the expected value of stimuli that guides adaptive behaviours⁴⁰. Furthermore, the habenula is found to preferentially interface with task-positive regions of the default mode network, including the mid- to posterior-PCC, potentially suggesting a role of the habenula in self-oriented processes engaged by external activities³⁶. In rats, synchronised activity between the habenula and the hippocampus is thought to contribute to the encoding of contextual information required for mnemonic processes^{41, 42}. However, a mechanistic account of the habenula's influence over these regions to support the processing of negative self-related cognitions remains undefined."*

The reviewer is correct in stating the potential relevance of the insula in processing negatively valenced information, which would be of interest for future studies. We have highlighted this as a future direction, with reference to the inherent constraint of DCM as quoted below.

Lines 345-350: “Moreover, the successful inversion of DCM requires an efficient (small-scale) model structure with region selection based on a priori network hypotheses⁴⁸. To more comprehensively characterise habenula involvement in negative self-cognitions, future studies could expand the current model by considering additional regions, such as the insula, which is functionally connected to the habenula³⁸, and the disruption of which has been implicated across psychiatric disorders⁹⁰.”

36. Ely BA, Stern ER, Kim J-w, Gabbay V, Xu J. Detailed mapping of human habenula resting-state functional connectivity. *Neuroimage* **200**, 621-634 (2019).
38. Torrisi S, Nord CL, Balderston NL, Roiser JP, Grillon C, Ernst M. Resting state connectivity of the human habenula at ultra-high field. *Neuroimage* **147**, 872-879 (2017).
39. Ely BA, Xu J, Goodman WK, Lapidus KA, Gabbay V, Stern ER. Resting-state functional connectivity of the human habenula in healthy individuals: Associations with subclinical depression. *Hum. Brain Mapp.* **37**, 2369-2384 (2016).
40. Rolls ET. The roles of the orbitofrontal cortex via the habenula in non-reward and depression, and in the responses of serotonin and dopamine neurons. *Neurosci. Biobehav. Rev.* **75**, 331-334 (2017).
41. Goutagny R, et al. Interactions between the lateral habenula and the hippocampus: Implication for spatial memory processes. *Neuropsychopharmacology* **38**, 2418-2426 (2013).
42. Mathis V, Lecourtier L. Role of the lateral habenula in memory through online processing of information. *Pharmacol. Biochem. Behav.* **162**, 69-78 (2017).
48. Friston KJ, et al. Dynamic causal modelling revisited. *Neuroimage* **199**, 730-744 (2019).
90. Nord CL, Lawson RP, Dalgleish T. Disrupted dorsal mid-insula activation during interoception across psychiatric disorders. *Am. J. Psychiatry* **178**, 761-770 (2021).

2. Except for processes known to be strongly lateralized (e.g. language processing), I tend to be skeptical of asymmetric brain models/interpretations. I appreciated that the authors used a bilateral Hb ROI for functional analyses, and I understand that the selection of right PCC/hippocampus/OFC ROIs was driven by the stronger activation observed in these regions during the task, but the supplementary figures appear to show at least some left-hemisphere activation in these regions as well. The authors should repeat their DCM analysis including both left and right hemisphere ROIs to clarify whether the effects described in this paper are truly restricted to the right hemisphere, are weaker but still apparent in the left hemisphere, or perhaps whether some more complex interaction between hemispheres is at work. Given the procedure for defining subject-level activation peaks, this will likely reduce the overall power relative to the right-hemisphere-only version, so the authors may deem it more appropriate to present this as a supplementary analysis/result. If such an analysis is not feasible for this dataset, they should state clearly why this is the case.

Reply: We thank the reviewer for this thoughtful suggestion. The reviewer correctly noted that the pOFC and the hippocampus showed bilateral activation during the repeating of negative self-cognitions compared to restructuring, and that these effects were stronger in the right hemisphere. The PCC had only right-lateralised activity under our task contrast. While these results informed our right-lateralised model node selection, we agree with the reviewer that we cannot rule out potential effects of habenula effective connectivity in the left hemisphere.

As such, we conducted an additional exploratory DCM analysis using a left-lateralised model to examine potential functional lateralisation of habenula effective connectivity during the processing of negative self-cognitions in the discovery sample. Results from this analysis are now summarised in the main text Results section, as follows. Timeseries extraction and complete parameter estimates are reported in the Supplementary Methods and Supplementary Table 4, respectively.

Lines 183-193: *“Considering the bilateral activation of the hippocampus and the OFC during the repeating of negative self-cognitions, we conducted an exploratory DCM analysis using a left-lateralised model consisting of the bilateral habenula, left PCC, left hippocampus, and the left pOFC following identical model specification and inversion procedures described above. Detailed steps for timeseries extraction and the complete Bayesian model-averaged parameter estimates are reported in the Supplementary Information. In short, the left-lateralised model revealed broadly convergent connectivity patterns as the right-lateralised model, recapitulating the excitatory connectivity from the habenula to the PCC and pOFC during cognitive restructuring (Supplementary Table 4). Additional excitatory modulatory effects were observed for the habenula-to-left hippocampus and PCC-to-habenula pathways during restructuring. The habenula-to-PCC excitatory connectivity during the repeating of self-cognitions was not observed in the left-lateralised model.”*

The additional excitatory connectivity observed on the habenula-to-left hippocampus pathway may reflect potential functional lateralisation of episodic memory processing required during the restructuring of negative self-cognitions. We have expanded our Discussion on the habenula-hippocampus findings to elaborate upon this.

Lines 319-332: *“Contrary to our hypotheses, we did not find sufficient evidence to support modulation in connectivity between the habenula and the right hippocampus during negative self-cognition processing. However, the exploratory left-lateralised model revealed excitatory modulation from the habenula to the left hippocampus during the challenge condition, suggesting that the two hippocampi may be differentially engaged during the restructuring of negative self-cognitions. This is broadly consistent with prior literature showing lateralised hippocampal function, such that the left hippocampus primarily encodes egocentric information and spatiotemporal association that constitute components of episodic memory, whereas the right hippocampus contributes to contextual topographic representations crucial for spatial navigation^{85, 86, 87}. In early phases of learning, both hippocampi have been shown to coactivate, reflecting the novelty of a given situation and supplying complementary representations that support different aspects of memory formation⁸⁶. The bilateral hippocampal activation observed here may reflect the effect of exposure to new self-cognition statements in each trial, while connectivity between the habenula and the left hippocampus supports episodic memory processes required for cognitive restructuring drawing upon personal history.”*

Lastly, we acknowledge that our DCM analyses cannot address potential inter-hemispheric interactions given the limitation in the number of nodes and connections that can be feasibly include in a single DCM model. As included in the response to the reviewer’s first comment, we have proposed that expanded models should be explored in future studies.

85. Nemati SS, Sadeghi L, Dehghan G, Sheibani N. Lateralization of the hippocampus: A review of molecular, functional, and physiological properties in health and disease. *Behav. Brain Res.* **454**, 114657 (2023).

86. Iglói K, Doeller CF, Berthoz A, Rondi-Reig L, Burgess N. Lateralized human hippocampal activity predicts navigation based on sequence or place memory. *Proc. Natl. Acad. Sci. U. S. A.* **107**, 14466-14471 (2010).

87. Miller J, et al. Lateralized hippocampal oscillations underlie distinct aspects of human spatial memory and navigation. *Nat. Commun.* **9**, 2423 (2018).

Reviewer #2 (Remarks to the Author):

Thank you for the opportunity to review the manuscript titled "Habenula neural circuitry drives negative self-cognitions." This study is a novel and significant contribution, leveraging 7T fMRI and dynamic causal modeling (DCM) to investigate the role of the habenula in processing and restructuring negative self-cognitions. The work makes meaningful advancements in understanding the habenula's involvement in higher-order cognitive processes and its interactions with key brain regions such as the posterior cingulate cortex (PCC) and posterior orbitofrontal cortex (pOFC). While the study is innovative and well-executed, there are areas for further clarification and improvement, particularly concerning the title, the depth of behavioral analysis, and the broader clinical implications. Below, I outline my detailed evaluation:

1. Title: Potential Overstatement

The title suggests a deterministic role of the habenula in driving negative self-cognitions. However, the findings primarily indicate that the habenula modulates connectivity with the PCC and pOFC during specific cognitive processes. The causal mechanisms underlying negative self-cognition remain to be fully established.

Reply: We thank the reviewer for their feedback and agree that the former title may run the risk of implying the habenula network as the sole driver of negative self-cognitions. We have retitled the manuscript "*Habenula contributions to negative self-cognitions*" to more accurately represent our findings.

2. Limited Connection to Clinical Pathology

While the study provides valuable insights into habenula-centric networks in healthy individuals, it lacks direct links to pathological conditions like depression, where negative self-cognition is prominent. The authors could further discuss how these findings might be extended to clinical populations. For example, could resting-state functional connectivity or spectral DCM in patients with depression validate the habenula's role in pathological negative cognition? If the discovery sample includes resting-state fMRI, it would also be interesting to test whether the spectral DCM of the habenula and PCC shows a pattern consistent with the task-based findings, which would strengthen the study's conclusions.

Reply: We appreciate the reviewer's comment. The reviewer rightly noted that the present study sought to establish a normative understanding of habenula function during negative self-cognition processing in participants not experiencing psychopathology. We concur that negative self-cognitions are thought to be more entrenched and prominent in people with a psychiatric disorder (e.g., depression), and it would be insightful to extend the current habenula model to clinical populations. We have expanded the section on study limitations and future directions to acknowledge this and have proposed potential research questions for future studies based on the reviewer's suggestions.

Lines 339-345: "*While the present study focused on establishing a normative role of the habenula in negative self-cognition processing, future studies would benefit from applying our habenula-centric model in populations characterised by maladaptive cognitions (e.g., people experiencing psychopathology)*⁶. It could also be important to examine whether altered habenula circuitry confers risk for mental ill-health. For example, if elevated habenula-PCC connectivity and/or dampened habenula-OFC connectivity may underpin negatively biased self-perception in depression and post-traumatic stress disorder^{88, 89}."

As suggested, we conducted an exploratory analysis applying spectral dynamic causal modelling (spDCM⁴) to the discovery sample's resting-state data (please see Supplementary Methods for details) to probe resting-state effective connectivity of the present habenula-centric network model. Methods and results for this exploratory analysis are reported here.

Volumes-of-interest for the spDCM analysis were extracted from identical ROIs as our task-based DCM. Precisely, representative timeseries was calculated as the principal eigenvariate of all voxels within the individualised habenula mask or 4 mm radius spheres centring the subject-specific activation peak identified for each of the other model regions (i.e., right PCC, hippocampus, pOFC) during the task-based analysis. Similar to the task-based model, the resting-state model assumed bidirectional endogenous connections of the habenula from and to the other regions, in addition to their self-inhibitory connections. Inter-regional connectivity is measured in hertz (Hz), with positive values indicating an excitatory effect and negative values suggesting an inhibitory effect. Intra-regional self-inhibitory connections are log-scaled, with positive and negative values suggesting increased and decreased self-inhibition, respectively. Group-level effects were summarised using Parametric Empirical Bayes (PEB).

As illustrated in the figure below, we found evidence (posterior probability [PP] > .95) for self-inhibitory connectivity within the habenula ($Ep = -0.25$ Hz, $PP = 1.00$), the PCC ($Ep = -0.36$ Hz, $PP = .99$) and the pOFC ($Ep = -0.62$ Hz, $PP = .98$). However, suprathreshold evidence for inter-regional effective connectivity in this resting-state model was not found.

This contrasts our task-based findings to suggest that the excitatory effect from the habenula to the PCC and the pOFC during the processing of negative self-cognitions may reflect a potentially distinct functional interaction during the active engagement with negative self-cognition statements. This coincides with past findings showing habenula's preferential engagement with task-positive regions of the default mode network, including the mid-to-posterior cingulate cortex, alongside the reward system, salience network, and sensorimotor areas to suggest the habenula's sensitivity to externally oriented activities^b. Likewise, as discussed in the main text, the habenula-pOFC connectivity may play a role in guiding adaptive response to negatively valenced stimuli, which is not active at rest. Although our aims did not include an examination of resting-state effective connectivity in the habenula network in this healthy cohort, we agree with the reviewer that this warrants examination in future studies with clinical populations, where potential habenula alterations could contribute to underlying changes in processing negative cognitions.

Although the lack of inter-regional resting-state effective connectivity may appear to diverge from our functional connectivity findings, the two measures are technically and conceptually distinct. spDCM characterises the directional influence that the neuronal activity endogenous to one brain region exerts on another (in the spectral domain), given a biophysical model of how neuronal processes translate to observable BOLD time series data^a. In contrast, functional connectivity is the non-directional correlation between time series data. Model parameters estimated by spDCM capture the cross-covariance between all neural measurements at all time lags, while functional connectivity most commonly include only the zero-lag correlation^c. Importantly, changes to a single effective connectivity parameter have been demonstrated to cause a global impact across all functional connectivity values^c. These differences preclude the direct comparison of resting-state connectivity findings based on the two methods.

Resting-state effective connectivity of the habenula network. The colour grid represents the estimated strength of effective connectivity for the model connections, thresholded at posterior probability $>.95$. The x- and y-axes denote the out- and in-put regions, respectively. The colour bar depicts the range of estimated parameter.

Ep posterior expectation, *Hb* habenula, *HC* hippocampus, *PCC* posterior cingulate cortex, *pOFC* posterior orbitofrontal cortex.

6. Wahl K, et al. Is repetitive negative thinking a transdiagnostic process? A comparison of key processes of RNT in depression, generalized anxiety disorder, obsessive-compulsive disorder, and community controls. *J. Behav. Ther. Exp. Psychiatry* **64**, 45-53 (2019).

91. Davey CG, Harrison BJ. The self on its axis: A framework for understanding depression. *Transl. Psychiatry* **12**, 23 (2022).

92. Agathos J, et al. Neuroimaging evidence of disturbed self-appraisal in posttraumatic stress disorder: A systematic review. *Psychiatry Res. Neuroimaging* **344**, 111888 (2024).

Additional references in the current response:

a. Friston KJ, Kahan J, Biswal B, Razi A. A DCM for resting state fMRI. *Neuroimage* **94**, 396-407 (2014).

b. Ely BA, Stern ER, Kim J-w, Gabbay V, Xu J. Detailed mapping of human habenula resting-state functional connectivity. *Neuroimage* **200**, 621-634 (2019)

c. Novelli L, Friston K, Razi A. Spectral dynamic causal modeling: A didactic introduction and its relationship with functional connectivity. *Netw. neurosci.* **8**, 178-202 (2024).

3. Behavioral Data Analysis

The manuscript could delve deeper into the behavioral data from the cognitive restructuring task, particularly decision-making during the "decide phase" (restructure vs. repeat). For example: Are there individual differences in decision patterns, and how might they correlate with habenula connectivity? Could modeling approaches, such as drift-diffusion models (DDM), provide insights into decision biases or latencies linked to negative cognition? By exploring these aspects, the connection between neural findings and their cognitive-behavioral implications would be significantly strengthened.

Reply: We thank the reviewer for their helpful suggestions. We have conducted additional analyses using available data to more thoroughly characterise the participants' decisional biases to either restructure or repeat the statements depending on their overall negative self-cognition endorsement. The results of these analyses are summarised here and reported in the Supplementary Information, as quoted below.

Supplementary Results, section *Decisional bias to restructure or repeat self-cognitions*: “Across both discovery and replication samples, the participants' choice to either restructure/challenge or repeat negative self-cognitions during the cognitive restructuring task appeared to depend on levels of negative self-cognition endorsement (Supplementary Fig. 6). Specifically, average negative self-cognition endorsement was significantly higher for statements that were repeated versus those that the participants restructured ($Mean_{difference} = -0.48$, $t_{98} = -6.70$, $P_{bonf.-corrected} < .001$, $d = -0.67$, $95\% CI = [-0.62, -0.33]$). Considering this, a decisional bias score was calculated for each participant as the difference in average endorsement ratings on statements that they chose to repeat versus restructure, with more positive values reflecting more pronounced choice tendencies to repeat statements that were more strongly endorsed. Participant bias score was significantly positively correlated with negative self-cognition endorsement ($r = .33$, $P_{bonf.-corrected} < .001$, $95\% CI = [0.14, 0.50]$), suggesting that those with higher overall levels of negative self-cognition endorsement are more likely to show a stronger decisional bias to repeat more personally relevant statements as opposed to restructure them. This was further tested via *k*-mean clustering, where we broadly identified two sub-groups of participants who significantly differed on both their bias score and overall negative self-cognition endorsement (bias scores: $Mean_{difference} = -0.82$, $t_{97} = -7.02$, $P_{bonf.-corrected} < .001$, $d = -1.41$, $95\% CI = [-1.05, -0.59]$; endorsement ratings: $Mean_{difference} = -1.43$, $t_{97} = -11.30$, $P_{bonf.-corrected} < .001$, $d = -2.28$, $95\% CI = [-1.69, -1.18]$).”

Supplementary Figure 6. Participant negative self-cognition endorsement and choice bias during the cognitive restructuring task

a The bar graph depicts the mean and standard deviation of participant endorsement ratings for negative self-cognition statements that were either restructured/challenged or repeated during the cognitive restructuring task. Mean endorsement scores were higher for repeated statements compared to challenged statements ($Mean_{difference} = -0.48$, $t_{98} = -6.70$, $P_{bonf.-corrected} < .001$, $d = -0.67$, $95\% CI = [-0.62, -0.33]$). **b** The distribution plot shows participant choice biases calculated as the difference in endorsement ratings between statements that were challenged or repeated. Higher scores indicate an increased tendency to repeat statements that were more highly endorsed. **c** The scatter plot depicts the association between participant choice biases during the cognitive restructuring task and pre-task negative self-cognition endorsement ratings. A significant positive association was identified between choice biases and endorsement ratings ($r = .33$, $P_{bonf.-corrected} < .001$, $95\% CI = [0.14, 0.50]$), indicating that participants with an overall higher negative self-cognition endorsement rating were more likely to

repeat the statements they more strongly endorsed. This was further tested using k-means clustering. The green cluster ($n = 53$) had significantly lower endorsement rating (Mean_{difference} = -1.43, $t_{97} = -11.30$, $P_{\text{bonf.-corrected}} < .001$, $d = -2.28$, 95% CI = [-1.69, -1.18]) and bias scores (Mean_{difference} = -0.82, $t_{97} = -7.02$, $P_{\text{bonf.-corrected}} < .001$, $d = -1.41$, 95% CI = [-1.05, -0.59]) compared to the orange cluster ($n = 46$).

4. Discussion of Limitations

The discussion section acknowledges the healthy sample as a limitation but could expand on how this affects the study's generalizability to clinical contexts. For example, it would be interesting to test whether a clinical sample shows an atypical habenula-based pattern that drives the pathology of negative cognition. Additionally, the lack of significant habenula-hippocampus connectivity findings deserves more attention. Is this due to task design limitations, sample characteristics, or other factors?

Reply: We agree with the reviewer that it would be of importance to assess how habenula effective connectivity may differ in clinical population to confer vulnerabilities to mental ill-health. Accordingly, we have acknowledged this in our study limitation and expanded our discussion on future directions to propose how potential habenula network alterations may be relevant to psychological symptoms in mood and stress-related disorders.

Lines 339-345: *“While the present study focused on establishing a normative role of the habenula in negative self-cognition processing, future studies would benefit from applying our habenula-centric model in populations characterised by maladaptive cognitions (e.g., people experiencing psychopathology)⁶. It could also be important to examine whether altered habenula circuitry confers risk for mental ill-health. For example, if elevated habenula-PCC connectivity and/or dampened habenula-OFC connectivity may underpin negatively biased self-perception in depression and post-traumatic stress disorder^{88, 89}.”*

We also appreciate the reviewer's comment on the need for more in-depth discussions regarding the habenula-hippocampus findings. We have substantiated the Discussion section on habenula-hippocampus connectivity with reference to the additional results from the exploratory left-lateralised model suggested by Reviewer 1, as included below.

Lines 319-332: *“Contrary to our hypotheses, we did not find sufficient evidence to support modulation in connectivity between the habenula and the right hippocampus during negative self-cognition processing. However, the exploratory left-lateralised model revealed excitatory modulation from the habenula to the left hippocampus during the challenge condition, suggesting that the two hippocampi may be differentially engaged during the restructuring of negative self-cognitions. This is broadly consistent with prior literature showing lateralised hippocampal function, such that the left hippocampus primarily encodes egocentric information and spatiotemporal association that constitute components of episodic memory, whereas the right hippocampus contributes to contextual topographic representations crucial for spatial navigation^{85, 86, 87}. In early phases of learning, both hippocampi have been shown to coactivate, reflecting the novelty of a given situation and supplying complementary representations that support different aspects of memory formation⁸⁶. The bilateral hippocampal activation observed here may reflect the effect of exposure to new self-cognition statements in each trial, while connectivity between the habenula and the left hippocampus supports episodic memory processes required for cognitive restructuring drawing upon personal history.”*

85. Nemati SS, Sadeghi L, Dehghan G, Sheibani N. Lateralization of the hippocampus: A review of molecular, functional, and physiological properties in health and disease. *Behav. Brain Res.* **454**, 114657 (2023).

86. Iglói K, Doeller CF, Berthoz A, Rondi-Reig L, Burgess N. Lateralized human hippocampal activity predicts navigation based on sequence or place memory. *Proc. Natl. Acad. Sci. U. S. A.* **107**, 14466-14471 (2010).

87. Miller J, et al. Lateralized hippocampal oscillations underlie distinct aspects of human spatial memory and navigation. *Nat. Commun.* **9**, 2423 (2018).

5. Clinical Implications

While the habenula's role in negative self-cognition is intriguing, the practical applications of these findings remain underexplored. It would be interesting to highlight the potential for neuromodulators interventions targeting the habenula to influence cognitive restructuring or alleviate maladaptive negative self-cognitions. The authors could also discuss the relevance of these findings for optimizing therapeutic techniques, such as CBT, by integrating neural insights.

Reply: We thank the reviewer for this suggestion, which have now been integrated into the study conclusion regarding the potential translational value of our findings. We highlight the possibility to leverage habenula effective connectivity to guide novel treatment approaches in addressing entrenched negative self-cognitions (e.g., neuromodulation, ketamine treatment for depression), as well as how habenula effective connectivity parameters may be used as predictors to treatment that relies on individual capacity for cognitive restructuring (e.g., cognitive behavioural therapy).

Lines 362-368: “*A mechanistic account of habenula functioning lays the foundation to examine neural vulnerabilities contributing to maladaptive thinking patterns, and whether habenula connectivity may be predictive of response to interventions that rely on individual capacity to restructure negative self-cognitions (e.g., cognitive behavioural therapy)⁹¹. Our results could also inform future work assessing the habenula's potential as a treatment target to alleviate entrenched negative self-cognitions that do not respond to conventional psychotherapy alone, such as via neuromodulatory interventions or novel pharmacological agents impacting habenula circuitry (e.g., ketamine)⁹².*”

94. Kung P-H, Davey CG, Harrison BJ, Jamieson AJ, Felmingham KL, Steward T. Frontoamygdalar effective connectivity in youth depression and treatment response. *Biol. Psychiatry* **94**, 959-968 (2023).

95. Dai Y, Harrison BJ, Davey CG, Steward T. Towards an expanded neurocognitive account of ketamine's rapid antidepressant effects. *Int. J. Neuropsychopharmacol.* **28**, pyaf010 (2025).

Reviewer #3 (Remarks to the Author):

This study exemplifies excellence in neuroimaging research, demonstrating a high level of methodological rigour and innovation. The use of high-magnetic field strength provides exceptional data quality. At the same time, the advanced application of Dynamic Causal Modeling (DCM) analyses showcases the authors' expertise in utilizing cutting-edge analytical tools. The inclusion of subcortical regions, often overlooked in similar studies, further enhances the depth and comprehensiveness of the work. Additionally, the innovative task design reflects a thoughtful approach to addressing a timely and significant research question. Including two well-powered datasets ensures the reliability and robustness of the findings, making the results highly impactful and generalizable.

The manuscript is very well written, with a clear and coherent structure. The figures are of excellent quality and effectively support the interpretation of the results. Despite the complexity of the analyses, the authors provide sufficient clarity for the study to be accessible to readers with varying levels of expertise. This work represents a significant advance in the field of neuroimaging and aligns perfectly with the high standards of this journal. In summary, this is a well-executed and timely contribution to the field of neuroimaging research. I strongly recommend this manuscript for publication, as it meets the journal's high standards and advances the state of the art in multiple dimensions.

While the manuscript is already of exceptional quality, I offer the following minor suggestions to enhance its clarity and impact further:

1. If applicable, including a table within the main text with the coordinates of the ROIs used in the DCM analysis would be beneficial. While these coordinates are reported in Table S4, it is unclear what specific regions were used as ROIs in that table.

Reply: We thank the reviewer for this suggestion. We have moved the table with group-level ROI centre coordinates from the Supplementary Information to the main text (Table 1) and edited the table header to clarify the specific regions included in both the discovery and replication models. This is also referenced in the Results section when introducing our model space, quoted below.

Lines 153-157: *“Based on these initial activation results ($n = 48$) and past neuroimaging evidence, a DCM network including the habenula, right PCC, right hippocampus, and right pOFC as regions-of-interest (model nodes; group-level centre coordinates reported in Table 1) was inverted for each participant to infer the modulatory influence of negative self-cognition processing on habenula effective connectivity (Fig. 2e-f).”*

2. The discussion of the task-based GLM results, reported in the supplementary material on page 3, could be moved to the main results section as this analysis forms the foundation for the subsequent DCM analysis. Additionally, these findings overlap significantly with activity related to reappraisal, which could be discussed further.

Reply: Thank you for the comment. We agree with the reviewer that the task-based GLM results are foundational for the DCM analysis. While we elected to report the complete GLM findings in the Supplementary Information due to space constraints of the main text and improve narrative clarity, we have now expanded the results section in the main text to further clarify activation laterality based on a related comment from Reviewer 1.

Lines 145-152: *“Mass-univariate general linear model (GLM) activation analysis revealed that the habenula had increased activity during the repeating of negative self-cognitions compared to restructuring, in addition to the right PCC, bilateral hippocampus, and the bilateral pOFC (Fig 2; Supplementary Fig. 1 & Table 2). Of note, neural activation elicited by the repeating of negative self-*

cognitions was more pronounced in the right hemisphere in both the hippocampus and the pOFC compared to the left hemisphere. As illustrated in Fig. 2d, habenula response increased during both the repeating and restructuring of negative self-cognitions relative to rest. Complete GLM activation results are reported in Supplementary Table 2.”

We have also added the following text to the Discussion section to elaborate upon the neural activation results relating to cognitive restructuring. The reviewer is correct to note the similarities of these results with cognitive reappraisal, which is highlighted in our discussion.

Lines 275-285: *“Restructuring negative self-cognitions not only requires the sustaining of complex self-concepts, subserved by the default mode network⁶⁸, but also the manipulation of self-representations, which has been shown to involve frontostriatal valuation and cognitive control circuits^{50, 69}. Our observed widespread activation in the lateral PFC control regions and the pre-supplementary motor area extending across the dorsomedial PFC, as well as the caudate and the thalamus, during the restructuring of negative self-cognitions compared to repeating further supports this and is largely consistent with past studies on cognitive emotion regulation highlighting the interaction between frontoparietal and subcortical structures when downregulating negative affect^{62, 70}. Under this framework, the PFC is thought to carry out reflective and contextual appraisals/evaluations, while subcortical and brainstem regions supply more basic information defining the affective state (e.g., salience, valence, physiological response)⁷¹.”*

50. Steward T, et al. A thalamo-centric neural signature for restructuring negative self-beliefs. *Mol. Psychiatry* **27**, 1611-1617 (2022).

62. Steward T, et al. Dynamic neural interactions supporting the cognitive reappraisal of emotion. *Cereb. Cortex* **00**, 1-13 (2020).

68. Davey CG, Pujol J, Harrison BJ. Mapping the self in the brain's default mode network. *Neuroimage* **132**, 390-397 (2016).

69. Agathos J, et al. Differential engagement of the posterior cingulate cortex during cognitive restructuring of negative self- and social beliefs. *Soc. Cogn. Affect. Neurosci.* **18**, (2023).

70. Dixon ML, Moodie CA, Goldin PR, Farb N, Heimberg RG, Gross JJ. Emotion regulation in social anxiety disorder: Reappraisal and acceptance of negative self-beliefs. *Biol. Psychiatry Cogn. Neurosci. Neuroimaging* **5**, 119-129 (2020).

71. Dixon ML, Thiruchselvam R, Todd R, Christoff K. Emotion and the prefrontal cortex: An integrative review. *Psychol. Bull.* **143**, 1033-1081 (2017).

3. Figure 2: The labels in the legend of Fig. 2A, 2B, 2C, and 2D are tiny—please increase the font size. The coordinates in Fig. 2C are challenging to read, and highlighting the habenula activity with a circle could enhance clarity.

Reply: We appreciate the reviewer’s feedback and have increased the font size for the labels in the legend of Fig. 2. We also provide high-resolution figure files with the submission, which can be scaled to optimise readability in the publication. We also added yellow circles highlighting the habenula activation cluster for Fig. 2c to improve clarity as suggested.

4. Figures 2F, 3A, and 4A: The color of the pOFC in the brain images does not match the legend. Please ensure that the colours are consistent throughout.

Reply: We thank the reviewer for bringing the formatting inconsistency to our attention. We have updated Fig. 2, 3, and 4 to ensure consistent colouring for the pOFC across the images and legends.

5. Figure 4B: The abbreviation “CHAL” should be explained in the figure legend for clarity.

Reply: We agree with the reviewer that abbreviations for the task conditions should be clearly explained. We have clarified this in the figure caption for Fig. 4b, as quoted below, as well as all figure referencing the task conditions throughout the manuscript.

Lines 1145-1154: *“Solid arrows represent intrinsic connectivity while dashed arrows illustrate modulatory effects by the task conditions – the restructuring (CHAL condition) or repeating (REP condition) of negative self-cognitions. Red arrows show excitatory intrinsic connectivity or positive modulatory effects. Blue arrows indicate inhibitory intrinsic connectivity or negative task-induced modulation. b Changes in habenula effective connectivity are plotted for connectivity showing significant task-induced modulation. The first column quantifies the average effective connectivity throughout the task for each pathway (intrinsic connectivity; A-matrix), representing the context-independent influence from the habenula to the PCC (blue) and the pOFC (green). The second and third column show the net effective connectivity of each pathway under the repeat (REP) and challenge (CHAL) conditions.”*

Reviewer #4 (Remarks to the Author):

Response: We thank the reviewer for their contributions to the peer review report.